# The Role of Receptor Tyrosine Kinases in Lassa Virus Cell Entry

**DOI:** 10.3390/v12080857

**Published:** 2020-08-06

**Authors:** Chiara Fedeli, Hector Moreno, Stefan Kunz

**Affiliations:** Institute of Microbiology, Lausanne University Hospital and University of Lausanne, 1011 Lausanne, Switzerland; chiara.fedeli1984@gmail.com

**Keywords:** Lassa virus, tyrosine kinase receptors, Axl, HGFR, alternative viral receptors, macropinocytosis, antiviral agents

## Abstract

The zoonotic Old World mammarenavirus Lassa (LASV) causes severe hemorrhagic fever with high mortality and morbidity in humans in endemic regions. The development of effective strategies to combat LASV infections is of high priority, given the lack of a licensed vaccine and restriction on available treatment to off-label use of ribavirin. A better understanding of the fundamental aspects of the virus’s life cycle would help to improve the development of novel therapeutic approaches. Host cell entry and restriction factors represent major barriers for emerging viruses and are promising targets for therapeutic intervention. In addition to the LASV main receptor, the extracellular matrix molecule dystroglycan (DG), the phosphatidylserine-binding receptors of the Tyro3/Axl/Mer (TAM), and T cell immunoglobulin and mucin receptor (TIM) families are potential alternative receptors of LASV infection. Therefore, the relative contributions of candidate receptors to LASV entry into a particular human cell type are a complex function of receptor expression and functional DG availability. Here, we describe the role of two receptor tyrosine kinases (RTKs), Axl and hepatocyte growth factor receptor (HGFR), in the presence and absence of glycosylated DG for LASV entry. We found that both RTKs participated in the macropinocytosis-related LASV entry and, regardless of the presence or absence of functional DG, their inhibition resulted in a significant antiviral effect.

## 1. Introduction

The Old World (OW) mammarenavirus Lassa virus (LASV) is a zoonotic pathogen and the etiologic agent of Lassa hemorrhagic fever (LHF), a severe disease with a high case fatality rate in humans [1]. Endemic in large parts of Western Africa, LASV currently threatens over 200 million people causing a heavy disease burden [2,3]. The only available treatment against LHF is the off-label administration of ribavirin (Rib), which is frequently associated with significant side effects [4]. Because of the serious public health concern, lack of a licensed vaccine, and limited therapeutic options, the World Health Organization considers LASV among the most important emerging pathogens [5]. The natural reservoirs of LASV are semi-domestic rodents of the *Mastomys* species and transmission of LASV occurs mainly via zoonosis by inhalation of aerosolized contaminated rodent excreta or consumption of contaminated food [1,6]. Human-to-human transmission can occur in nosocomial settings and is associated with a high case fatality rate [7]. Following early multiplication at the site of entry, LASV can spread systemically and cause severe disseminated infection, characterized by high viral loads in serum and organs [8]. Survivors develop a timely adaptive antiviral immune response and clear the virus. Severe cases have shown marked virus-induced immunosuppression, resulting in uncontrolled viral multiplication, systemic shock, and death. Serum viral load, early in the disease, is of predictive value for disease outcome [9]. Antiviral drugs capable of limiting viral spread could provide a patient’s immune system a window of opportunity to develop a protective antiviral response. Thus, targeting viral entry to block the virus before it takes control of the host cell appears to be a promising strategy for therapeutic intervention.

Mammarenaviruses are enveloped negative-strand RNA viruses with a non-lytic life cycle confined to the cytosol [10]. The mammarenavirus genome is comprised of two RNA segments with ambisense coding strategy. A small (S) RNA segment encodes the envelope glycoprotein precursor (GPC) and the nucleoprotein (NP), whereas a large (L) segment encodes the matrix protein (Z) and the viral RNA-dependent RNA polymerase (L). Maturation of the GPC precursor involves proteolytic processing by cellular signal peptidases and the proprotein convertase subtilisin kexin isozyme-1/site-1 protease (SKI-1/S1P), yielding a retained stable signal peptide (SSP), the N-terminal attachment protein GP1, and the membrane-anchored fusion protein GP2 [11]. Trimers of the non-covalently associated SSP/GP1/GP2 subunits form the mature GP and decorate the virion surface [11,12,13].

The main cellular human receptor for OW mammarenaviruses LASV and lymphocytic choriominingitis virus (LCMV) is the ubiquitously expressed extracellular matrix (ECM) receptor dystroglycan (DG) [14,15]. DG is evolutionary highly conserved and provides a molecular link between the ECM and the cytoskeleton [16]. The DG precursor undergoes autoproteolytic processing, yielding the peripheral α-DG that binds ECM proteins and the transmembrane β-DG, which anchors the complex to the actin cytoskeleton. During biosynthesis, α-DG is subject to remarkably complex O-glycosylation, including attachment of [Xyl-α1-GlcA-3-β1-3] copolymers by the glycosyltransferase like-acetylglucosaminyltransferase (LARGE) [17,18]. These LARGE-derived O-linked glycan polymers, known as matriglycan, are crucial for high affinity binding of mammarenaviral GP to ECM proteins [19,20,21,22,23,24,25]. Most cell types in the mammalian body express the DG core protein, whereas functional glycosylation of α-DG is subject to tight tissue-specific regulation, making DG a “tunable” receptor [19]. After binding to DG, LASV undergoes endocytosis via a not yet clearly understood macropinocytosis-related pathway that critically depends on sodium proton exchangers (NHE) and actin [26,27]. Receptor-mediated endocytosis of LASV is followed by delivery to late endosomes, involving the endosomal sorting complexes required for transport (ESCRT) [28]. At the late endosome, acidic pH induces conformational changes in LASV GP1, resulting in dissociation from DG and engagement of the lysosomal-associated membrane protein-1 (LAMP-1), which triggers membrane fusion [29,30,31]. In the absence of LAMP-1, LASV endosomal escape takes place at later and more acidified endosomal compartments [32]. The capacity to use LAMP-1 as an endosomal entry factor is unique for LASV and not shared by other mammarenaviruses [33].

Post-mortem examination of fatal human LASV cases have revealed that the tissue tropism of LASV did not always correlate with the functional glycosylation of DG [8,34]. Productive viral entry into cells lacking functional DG have been explained by the presence of alternative virus receptors, including phosphatidylserine (PS) receptors of the Tyro3/Axl/Mer (TAM) and the T cell immunoglobulin and mucin receptor (TIM) families, as well as C-type lectins [35,36,37,38]. The receptors, TAM and TIM, both participate in relevant immunological functions, including their evolutionary ancient role in removal of apoptotic debris, and promotion of the activation and proliferation of T cells [39,40,41]. The TAM receptors play a crucial role in the regulation of the host type interferon (IFN)-I response, which represents a major branch of innate antiviral immunity [42,43,44]. Conserved PS receptors of the TAM/TIM families can mediate cell entry of a wide range of enveloped viruses via “apoptotic mimicry” [45]. Apoptotic mimicry consists of the utilization of TAM and TIM cellular receptors to mediate viral particle binding to the cell surface through recognition of PS exposed in viral membrane. Despite not directly involving viral GP interaction, viral particle architecture, GP abundance, and lipid composition of viral particles are factors that can determine PS accessibility to participating cellular receptors. Many emergent viruses benefit from apoptotic mimicry during zoonotic events and adaptation to new hosts [36,46,47,48,49,50,51,52,53]. The TAM family member Axl is a receptor tyrosine kinase (RTK) expressed in many human cell types and utilized by LASV in vivo [54], contributing to its entry [36,37]. However, due to the high affinity binding between available DG and LASV_GP_, the relative contribution of Axl RTK to LASV entry critically depends on the extent of functional glycosylation of DG. In cells lacking the glycosylated DG form, LASV viral particles cannot bind DG, and Axl RTK is able to promote viral entry acting as a functional receptor and mediating virus binding and internalization [36,37]. Moreover, Axl RTK signaling also promotes the LASV endocytic macropinocytosis-like pathway [36,37]. In human microvascular endothelial cells that express Axl RTK in the context of an under-glycosylated DG form, the two receptors, Axl RTK and DG, seem to cooperate [37]. Human epithelial cells lining the respiratory and digestive tracts, which are likely to be initial targets during zoonotic LASV transmission, express Axl RTK in combination with functionally glycosylated DG. Several lines of evidence support a crucial role of functionally glycosylated DG in productive LASV entry into human epithelial cells [25,27,36,55,56]. However, the exact role of Axl RTK in this context is currently unclear. It is conceivable that in epithelial cells, long DG-linked matriglycan chains reach above the glycocalix and capture free virus. This would make attachment of the virion to the PS receptors less relevant, a notion that has been supported, at least in part, by previous antibody perturbation studies [27,36,37]. However, this working model does not rule out a functional contribution of Axl RTK to DG-mediated LASV entry at steps upon early attachment or to DG-independent entry.

Cellular factors participating in LASV virion entry determine cell susceptibility and affect subsequent steps of viral attachment. Indeed, RTKs, which are well-known upstream regulatory factors of macropinocytosis, participate in entry of a range of viruses [45,50,57,58,59], and therefore could be important for LASV entry. The hepatocyte growth factor receptor (HGFR) is an RTK expressed in many cell types, including epithelial cells, neurons, hepatocytes, and hematopoietic cells [60], some of which are important targets of LASV in vivo [61]. Despite its main function as a receptor to hepatocyte mitogen [62], HGFR participates in the control of cell motility, angiogenesis, the immune response, cell differentiation, and anti-apoptosis [63]. Furthermore, HFGR is involved in macropinocytosis [64] and its inhibition with EMD1204063 (EMD), an inhibitor currently tested in clinical trials for treatment of solid tumors [65], resulted in an antiviral effect against LASV infection [27].

Since work with replication-competent LASV is restricted to BSL4 high-containment facilities, for the present study, we employed a recombinant LCMV expressing the envelope (GP) LASV_GP_ (rLCMV/LASV_GP_) to investigate the molecular mechanisms underlying LASV entry. The rLCMV/LASV_GP_ chimera has been extensively used by others and us and has faithfully recapitulated LASV cell entry and tropism in vitro and in vivo at BSL2 facilities [26,27,28,37,66,67,68]. Here, we investigated the role of two RTK receptors, Axl and HGFR, on LASV entry in the presence and absence of functional DG. We found that both RTKs participated in early steps of LASV entry, altering macropinosome trafficking. Axl and HGFR inhibition resulted in antiviral effects through perturbation of cytoplasmic trafficking of virus-containing vesicles.

## 2. Materials and Methods

### 2.1. Antibodies and Reagents

Monoclonal antibodies to alpha-dystroglycan (IIH6) for glycosylated DG have been described previously [69]. Purified polyclonal goat IgG anti human Axl (#AF154) and anti-human HGFR (for IFA, #AF276) were purchased from R&D Systems (Minneapolis, MN, USA). Monoclonal 113 anti-LCMV NP antibody has been described previously [70]. For Western blot, anti-human HGFR and anti-human phospho-HGFR (10/2017 and 08/2017, respectively) were purchased from Cell Signalling (Leiden, The Netherlands). Horseradish peroxidase (HRP)-conjugated secondary antibodies were obtained from Dako, and phycoerythrin (PE)-conjugated secondary antibodies from Biolegend. The CellTiter-Glo assay system was purchased from Promega (Madison, WI, USA). Dextran, Alexa Fluor^TM^ 488 (10 kDa), and NucBlue^TM^ Fixed cells ReadyProbes^TM^ Reagent were purchased from Thermo Fisher (Waltham, MA, USA). R428 was obtained from Selleckem (Houston, TX, USA). EMD, EIPA, Rib, BafA1, and HGF were purchased from Sigma (St. Louis, MO, USA). 

### 2.2. Cell Culture

Human lung carcinoma alveolar epithelial A549 cells, the human fibrosarcoma cell line HT-1080, and the human hepatoma cell line Huh7 were originally obtained from ATCC and were maintained at 37 °C, 5% CO_2_, in Dulbecco’s modified Eagle medium containing high glucose (4.5 mg/L) and GlutaMAX™ (DMEM. Gibco BRL, Reinach, Switzerland), supplemented with 10% fetal calf serum (FCS) and penicillin (100 U/mL)/streptomycin (0.1 mg/mL). The human umbilical vein endothelial cells (HUVEC) (CC-2517) were purchased from Clonetics/Lonza as cryopreserved specimens and cultured according to the manufacturer’s protocol using the endothelial cell growth medium (EGM) BulletKit. The baby kidney hamster (BHK)-21 cells were maintained at 37 °C, 5% CO_2_, in DMEM containing high glucose (4.5 mg/L) and GlutaMAX™, supplemented with 5% FCS and penicillin (100 U/mL)/streptomycin (0.1 mg/mL), and nonessential amino acids (Gibco BRL, Reinach, Switzerland).

### 2.3. Viruses and Cell Infection

Production, growth, and titration of rLCMV/LASV_GP_, rLCMV-VSV_G_, and LCMV Arm 53b have been described elsewhere [68,71,72]. Recombinant AdV5-EGFP has been described in [73]. VSV Indiana strain and VSV-based pseudotype production have been described previously [74,75]. For viral infection, confluent cell monolayers were treated with drugs as detailed in the specific experiments, followed by infection with the indicated viruses, for 1 h, at 37 °C. Supernatants were removed and fresh DMEM containing 10% FCS and 20 mM NH_4_Cl was added. Infection of rLCMV/LASV_GP_ was quantified by detection of LCMV_NP_ in immunofluorescence assay (IFA) with mAb 113, as described in [66].

### 2.4. Flow Cytometry Analysis

Cells were seeded in 24-well plates at 1 × 10^5^ cells per well. After 24 h, cells were gently detached with a cell scraper and resuspended in fluorescence-activated cell sorting (FACS) buffer (1% (*v*/*v*) FCS solution, 0.1% (*w*/*v*) sodium azide, PBS). Cells were stained with primary monoclonal anti-αDG antibody (IIH6), polyclonal goat anti-Axl antibody (#AF154), or polyclonal goat anti-HGFR antibody (#AF276) (diluted 1:50 in FACS buffer), for 1 h, at 4 °C. Cells were washed twice with cold FACS buffer and labeled with secondary antibodies conjugated to PE for 45 min at 4 °C, in the dark. Cells were washed twice with cold FACS buffer and fixed with CellFix solution for 10 min, at room temperature, in the dark, and resuspended in PBS. Flow cytometry was performed using a BD Accurri^TM^ C6 Flow Cytometer (BD Bioscience, San Jose, CA, USA), collecting 10,000 events.

### 2.5. TCEP Assay

rLCMV/LASV_GP_ was produced in BHK21 cells and purified by ultracentrifugation as described in [76]. Then, the virus was resuspended in PBS and labeled with the thiol-cleavable reagent NHS-SS-biotin (Pierce), as described in [55]. As a cleavage control, we added the membrane-impermeable reducing agent Tris(2-carboxyethyl)phosphine (TCEP) (10 mM) (Pierce) for 30 min, which resulted in a loss of >95% of biotin label. As described previously [77], A549 cells confluent monolayers were preincubated with R428 (2 µM) or EIPA (20 µM), or left untreated for 30 min, at 37 °C, and then washed twice with cold HBSS and cooled down to 4 °C on ice, for 5 min. Cold solution containing NHS-SS-biotinylated rLCMV/LASV_GP_ (100 PFU/cell) and EIPA or R428 in HBSS was added and incubated on ice, for 1 h. After washing with cold HBSS, and to allow internalization, cells were shifted to 37 °C for the indicated times. Then, cells were cooled down (4 °C) and TCEP (15 mM) in 50 mM HEPES, pH 7.5, 150 mM NaCl, 1 mM CaCl_2_, and 1 mM MgCl_2_ were added and incubated for 30 min, on ice. Excess of TCEP was removed by washing three times with cold HBSS. The remaining TCEP was quenched with 100 mM iodoacetamide for 10 min, and cells were lysed immediately. Then, 10^7^ PFU of purified, unlabeled rLCMV/LASV_GP_ were added to cleared lysates as carrier and IP was performed with mAb 83.6 to LASV GP2 immobilized on Sepharose 4B as described in [77]. Complexes were separated by nonreducing SDS-PAGE and biotinylated LASV GP2 was detected by Western blotting with HRP-conjugated streptavidin using ECL for detection.

### 2.6. RNAi

For Axl knockdown, 4 × 10^4^ cells/well were seeded overnight in a 96-well plate, and transfected with All Star Scramble RNA (Qiagen, Hilden, Germany) or with On-Target plus SMARTpool for Axl (L-003104-00-0005) (Thermo Scientific Dharmacon, Lafayette, CO, USA) at a final concentration of 200 nM, using Lipofectamine RNAiMax (Invitrogen, Paisley, UK). After 48 h, cells were infected with AdV-EGFP, VSVΔG-EBOV_GP_ or VSVΔG-LASV_GP_ at MOI 0.01 PFU/cell. The effective knockdown of Axl was verified by Western blot. 

### 2.7. Immunoblotting

Proteins were separated by SDS-PAGE and transferred to nitrocellulose. After blocking in 5% (*w*/*v*) skim milk in PBS, 0.1% (*v*/*v*) Tween-20 (PBST), nitrocellulose membranes were incubated with primary antibody (1–10 µg/mL) in 5% (*w*/*v*) skim milk, PBST, in cold, for 12–16 h. Then, membranes were washed 3 times with PBST, and HRP-coupled secondary antibodies were applied 1:5000 in PBST (1 h, room temperature). After washing in PBST, blots were developed by enhanced chemiluminescence (ECL) using a LiteABlot kit (EuroClone, Pero, Italy). Signals were acquired by ImageQuant LAS 4000Mini (GE Healthcare Lifesciences, Glattbrugg, Switzerland). 

### 2.8. RNA Isolation and RT-qPCR

RNA isolation and RT-qPCR were performed, as described in [78]. Briefly, total RNA was isolated with a NucleoSpin RNA kit (Macherey-Nagel, Oensingen, Switzerland), eluted in 60 µL of nuclease-free water and RNA concentration was determined with a Qubit 4.0 (Thermofisher, Reinach, Switzerland) device. For reverse transcription reaction, we used 0.5 µg of total RNA, with a high-capacity cDNA reverse transcription kit from Applied Biosystems (Foster City, CA, USA). Human IFN-β and GAPDH transcripts were detected with specific primers and probes (Hs01077958_s1/FAM and Hs99999905_m1/VIC, respectively, Applied Biosystems, Reinach, Switzerland). qPCR was carried out with a StepOne qPCR system (Applied Biosystems), and IFN-β gene expression levels relative to GAPDH were determined according to the 2^−ΔΔ^*^C^**^t^* method (where *C**_t_* is the threshold cycle) [79].

### 2.9. Confocal Microscopy

To perform colocalization studies of membrane receptors, A549 cells were seeded onto sterile 12 mm glass cover slips at a density of 7 × 10^4^ cells/well. The day after, cells were washed with ice cold HBSS and incubated with HBSS 1% FCS for 20 min, on ice. Then, cells were stained with primary antibodies (mouse monoclonal IgM IIH6 anti-human α-DG, 1:100; goat polyclonal anti-human Axl, 1:100; and anti-human HGFR #AF276, 1:500) diluted in HBSS 1% FCS, for 1.5 h, in the cold. After 2 washes with ice cold HBSS, 1% FCS, cells were labeled with secondary antibodies (Alexa Fluor^®^ 594 AffiniPure F(ab’)2 Fragment Donkey Anti-Mouse IgM (H+L), 1:300 for DG detection; Alexa Fluor^TM^ 594 Donkey anti-Goat IgG (H+L), 1:300 for Axl and HGFR detection; for 1 h on ice, in the dark. Then, cells were washed twice with PBS and fixed with 2% paraformaldehyde/0.1% glutaraldehyde in PBS for 30 min at 4 °C, and 15 min at room temperature. Nuclei were stained with NucBlue^TM^ Fixed Cell ReadyProbes^TM^ Reagent and, finally, cells were coverslipped with Mowiol. Image acquisition was performed with a Zeiss LSM780 Quasar confocal microscope. Multiplier gain for each channel was adjusted to minimize background noise and saturated pixels. To quantify colocalization, Pearson’s coefficient was calculated on randomly taken, single confocal plane images, applying the Fiji Coloc2 Analysis plugin. Double positive puncta (for NP and Axl, or HGFR colocalization) was performed using Fiji software. Using the same threshold, images were converted to positive and negative signal and double positive puncta were quantified. The percentage of positive puncta is relative to the total number of NP positive puncta. 

### 2.10. Dextran Uptake Assay

A549 cells were seeded onto sterile 12 mm glass cover slips at a density of 7 × 10^4^ cells/well, incubated for 24 h and treated for 30 min with the indicated inhibitors, and subsequently, with Dextran, Alexa Fluor^TM^ 488 (10 kDa) at 0.25 mg/mL for an additional 40 min, at 37 °C. Cells were washed twice, fixed with 2% *v*/*v* PFA, and stained with NucBlue^TM^ to visualize the nucleus. After mounting with Mowiol, cells were directly analyzed by confocal microscopy. Viral NP and Dextran, Alexa Fluor^TM^ 488 colocalization was measured, as described in [78].

## 3. Results

### 3.1. Axl Is Required for LASV Entry into Human Cells

Human epithelial cells of the respiratory tract are likely to be early targets during zoonotic transmission of LASV [8] and to express Axl RTK in the context of functionally glycosylated DG [27,38]. Since functional studies of LASV entry into human epithelial cells would require significant amounts of homogeneous tissue culture material, work with primary human small airway epithelial cells (SAEC) is challenging. As recently reported, SAEC and the immortalized human alveolar epithelial cell line A549 showed similar levels of productive rLCMV/LASV_GP_ infection [80], which correlated with similar expression patterns of functional glycosylated DG and the alternative receptors Axl RTK and T cell immunoglobulin and mucin domain (TIM)-1 [27,37,80]. Therefore, similar susceptibility and candidate receptor expression make A549 cells a suitable model for functional studies of human respiratory epithelial cells, and expression of fully glycosylated DG and Axl in A549 cells has been previously confirmed in our laboratory [37] (Appendix A). LASV infection also perturbs endothelial function [81,82] and human umbilical cord microvascular endothelial cells (HUVEC) display expression of Axl RTK, as well as under-glycosylated DG [37], as validated by our fluorescence-activated cell sorting (FACS) analysis (Figure 1A).

The human fibrosarcoma cell line, HT-1080, which is susceptible to rLCMV/LASV_GP_ infection, has previously been shown to robustly express Axl RTK but lacked functional DG [36,37], as confirmed by our analyses (Appendix A). To evaluate the role and relative contributions of Axl RTK in different functional DG contexts, we compared rLCMV/LASV_GP_ infection in A549, HUVEC, and HT-1080 cells. Because we failed to transiently express in 293T cells both wild type Axl RTK and its variant bearing loss-of-function mutations in the intracellular domain (Appendix A), we used the Axl RTK inhibitor R428 in A549, HT-1080, and HUVEC cells, which expressed endogenous Axl RTK [37]. R428 is the most specific available inhibitor of Axl RTK and strongly inhibits its phosphorylation (IC_50_ < 30 nM). It is a reversible inhibitor and shows high specificity for Axl over other RTKs in cell-based assays [83,84], without affecting Axl RTK expression (Figure 1B). Because of the potential compensatory effect or off-target effects, such as Tie-2, Ftl-1, Flt-3, Ret, or Abl [83], we chose a short time window of drug action, combined with drug washout, assuring micro-reversibility, and therefore minimized potential consequences of the R428 administration and allowed us to dissect the Axl RTK role during LASV entry [85] (Figure 1C). Moreover, R428 is currently in phase II clinical trials for human anticancer therapy and has a favorable toxicity profile, suggesting minimal off-target effects [84,86]. For the entry assay (Figure 1C), cells were pretreated for 30 min, at 37 °C, with the indicated inhibitor, cooled down to 4 °C, infected with rLCMV-LASV_GP_, and shifted to 37 °C, for 1 h, in the presence of the drug. Then, the cell media was replaced with fresh media containing 20 mM of ammonium chloride (NH_4_Cl) to prevent subsequent replication rounds. When added to cells, ammonium chloride instantly raised the endosomal pH, preventing subsequent acidic-dependent endosomal escape of viruses, without causing overall cytotoxicity [87,88]. After 16–20 h, cells were fixed, and infection was quantified by immunofluorescence assay (IFA) for LCMV NP. Our results showed that R428 reduced rLCMV/LASV_GP_ infectivity in a dose-dependent manner in A549, HUVEC, and HT-1080 cells expressing functional Axl RTK (Figure 1D and Appendix A). The IC_50_ values for R428 were <1 µM in the cell types tested, resulting in a therapeutic index (TI = CC_50_/IC_50_) >40. None of the tested R428 concentrations exerted significant cellular toxicity (Appendix A).

To further confirm a role for Axl RTK in LASV infection, we performed infections in 293T cells, which have undetectable endogenous Axl RTK levels, and in 293T cells that constitutively overexpress Axl RTK (293T/Axl) (Figure 1E). To control the functional overexpression of Axl, we used Ebola virus (EBOV), whose infection is known to be promoted by Axl RTK [48,49,50]. Given the high biosecurity required for handling replication-competent EBOV, we used a vesicular stomatitis virus (VSV)-based pseudotype platform, in which the original VSV_G_ had been exchanged with an EGFP reporter to allow monitoring of viral infection, and the heterologous EBOV_GP_ was provided from an expression vector. This VSV-based pseudotype platform has been shown in previous studies to be a reliable surrogate for entry of several viruses, including filoviruses [89]. Axl RTK overexpression in 293T cells resulted in a two-fold increase (*p* < 0.01) of rLCMV/LASV_GP_ infection and a six-fold increase in VSVΔG-EBOV_GP_ transduction (Figure 1F and Appendix A). In contrast, adenovirus 5 (AdV5) infection remained unaffected by Axl RTK expression in 293T cells. Furthermore, depletion of Axl RTK in A549 cells resulted in an approximately two-fold reduction of rLCMV/LASV_GP_ (*p* < 0.01) and VSVΔG-EBOV_GP_ relative infection and transduction, respectively, while AdV5 infection remained unaffected (Figure 1G,H and Appendix A). In summary, our data provide evidence that LASV entry partially depends on Axl RTK expression. 

### 3.2. Axl Participates in Early Steps of LASV Entry

To further validate the participation of Axl RTK during the early steps of rLCMV/LASV_GP_ infection, we administered R428 before, during, and after viral infection (time-of-addition assay, Figure 2A).

To ensure viral infection synchronization, precooled (4 °C) cells were infected with rLCMV-LASV_GP_ and shifted to 37 °C, for 1 h, followed by washing with a medium containing 20 mM NH_4_Cl, and incubation for 16–20 h, at 37 °C. Indicated inhibitors were added at different times relative to the moment of the infection (−0.5, 0, +1, +2, or +4 hpi), in order to discriminate between effects on the viral entry or viral transcription/replication, and kept throughout the experiment, until fixation and quantification by IFA. The NHE inhibitor 5-(*N*-ethyl-*N*-isopropyl)-amiloride (EIPA) was used as a control for LASV entry inhibition, and the replication inhibitor Rib was included as an inhibitor of post-entry steps of infection [26,90]. Irrespective of the extent of DG glycosylation, R428 inhibited rLCMV/LASV_GP_ infection in both cell lines when added before or during entry, similar to EIPA, whereas only mild antiviral effects were observed at later post-entry steps of infection (Figure 2B,C and Appendix A and [37]).As expected from its mechanism of action, Rib exerted a potent antiviral effect in this experimental setup, but it did not reduce rLCMV/LASV_GP_ entry when administered during entry steps (Appendix A).

Since LASV entry is a complex, multiprocess, we sought to discern possible roles of Axl RTK in virus attachment and endocytosis from later steps of viral entry. To this end, we employed an established virus uptake assay [55,77,91] to track virus attachment and internalization of virion particles. In this assay format, purified rLCMV/LASV_GP_ was labeled with the thiol-cleavable reagent NHS-SS-biotin, resulting in a biotin label that was sensitive to reducing agents without affecting infectivity (Figure 2D). The thiol-sensitive biotin label could be cleaved with the membrane-impermeable reducing agent Tris(2-carboxyethyl)phosphine (TCEP). This resulted in specific cleavage of the biotin label from the virus exposed to the extracellular space, whereas the biotin label on internalized virus particles becomes resistant, and therefore provided a sensitive tool to measure viral internalization, which was able to distinguish viruses attached to “open cups” and in closed vesicles still at the membrane proximity. The results showed that treatment with R428 inhibited virus internalization (Figure 2E). In agreement with its mechanism of action, EIPA also efficiently prevented virus internalization. Pretreatment with either R428 or EIPA did not affect virus attachment, evidenced by similar signals for biotinylated GP2 detected in the absence of TCEP (Figure 2E).

To further evaluate the possible effects of R428 in viral fusion or early post-fusion steps during rLCMV/LASV_GP_ infection, we synchronized virus escape from late endosomes with the drug treatment, as described in [92]. Briefly, for the post-entry assay (Figure 2F); precooled cells were incubated at 4 °C with rLCMV-LASV_GP_, for 1 h, allowing virus binding without internalization. Then, cells were washed to remove the unbound virus, and quickly shifted to 37 °C, for 5 min, to allow virus internalization. Then, cells were incubated for 45 min, at 37 °C, with a medium containing 20 mM NH_4_Cl, in order to prevent viral pH-dependent endosomal escape. Then, indicated inhibitors were added to the cells for 30 min, and kept for 4 h in the absence of NH_4_Cl, to test their effect on post-entry steps. Finally, NH_4_Cl was added for 16–20 h, to prevent secondary infections, followed by IFA for LCMV NP. A comparison of entry and post-entry assays (Figure 1C and Figure 2F, respectively) revealed that R428 did not inhibit viral fusion and endosomal escape of rLCMV/LASV_GP_ (Figure 2G and Appendix A) but interfered with early entry steps. This is in contrast to bafilomycin A1 (BafA1), an inhibitor of lysosomal proton pumps used as a positive control here, which suppressed productive viral infection in pre- and post-entry assays. The actin inhibitor cytochalasin D (CytoD), which interfered with actin dependent entry, did not prevent endosomal fusion and early replication of rLCMV/LASV_GP_, as expected (Figure 2G and Appendix A).

### 3.3. LASV Colocalizes with Glycosylated DG within the Plasma Membrane

Our findings provided evidence for Axl RTK participation in LASV entry in the presence or absence of fully functional DG. Therefore, we investigated a possible clustering of LASV with DG or Axl RTK during viral entry. Using confocal microscopy, in agreement with previous reports, we found that rLCMV/LASV_GP_ attached preferentially to DG clusters at the plasma membrane, which could serve as “docking sites” for free virus (Figure 3A). Nevertheless, rLCMV/LASV_GP_ and Axl RTK colocalization did not increase over time upon viral infection (Figure 3B). Contrary to the spotted pattern observed in DG, it is noteworthy that Axl RTK signal was diffuse and extended along the viral membrane, rendering comparisons between DG/NP and Axl RTK/NP colocalizations challenging. Then, we sought to investigate if LASV could induce transient DG-Axl RTK clustering during viral entry. The results showed no differences in colocalization of DG and Axl as comparing with uninfected and with infected cells at different time points after infection (Figure 3C).

Axl RTK participates in infection of several RNA viruses [48,51,52,93]. Interestingly, Axl signaling seems to contribute to virus internalization, and can also promotes later steps of viral multiplication [51,52,93]. Virus-induced activation of Axl RTK can promote replication by antagonizing IFN-I signaling, as illustrated by the flaviviruses dengue, West Nile, and Zika [52,93,94]. To address whether or not the observed contribution of Axl RTK to LASV increased entry was mediated by suppression or the IFN-I response, we performed infections in the absence or presence of R428 in immunocompetent A549 cells and monitored IFNβ mRNA levels, at 16 h post infection. The results showed that the presence of R428 did not affect IFNβ expression triggered by infection with LCMV, rLCMV-LASV_GP_, or VSV (Figure 4A and Appendix A).

To further investigate if Axl RTK signaling could suppress the IFN-I response during LASV infection in the absence of glycosylated DG, we performed rLCMV-LASV_GP_ infections with or without R428 in HT-1080 cells, lacking functional DG (Figure 1A). R428 did not affect IFNβ expression upon OW mammarenaviruses or VSV control infections (Figure 4B and Appendix A). Nevertheless, in uninfected HT-1080 cells, the administration of R428 slightly increased the IFNβ mRNA levels, suggesting a potential alteration of the innate immune response in the absence of other stimuli. In summary, R428 inhibitory effects on rLCMV-LASV_GP_ productive infection are not due to an increased IFN-I response.

HGFR is a RTK that is implicated in the entry of LASV and other viruses into human epithelial cells [27,59]; however, its exact role during viral entry is currently unknown. To extend our previous studies, we determined the contribution of HGFR to LASV cell entry. We compared rLCMV/LASV_GP_ infections in A549 cells, which express Axl and HGFR RTKs in the context of functional DG, with infections in HT-1080 cells, which likewise express Axl and HGFR RTKs but lack functional glycosylated DG (Appendix A and Figure 5A). As described for the entry assay (Figure 1C), we preincubated A549 and HT-1080 cells, for 30 min, at 37 °C, with increasing concentrations of EMD which is a potent HGFR kinase inhibitor with an IC_50_ of 3 nM [65], which did not affect HGFR expression (Figure 5B). Then, cells were cooled down to 4 °C, infected with rLCMV-LASV_GP_ in the presence of the drug, and shifted to 37 °C, for 1 h. The results showed that inhibition of HGFR by EMD reduced rLCMV/LASV_GP_ entry in a dose-dependent manner in both A549 cells (IC_50_ = 8.5 µM and CC_50_ > 90 µM) and HT-1080 cells (IC_50_ = 3.7 µM and CC_50_ > 90 µM) (Figure 5C and Appendix A), in the absence of overall cytotoxicity (Appendix A). 

To analyze the role of HGFR from a different angle, we first stimulated HGFR in A549 cells with exogenous HGF (50 nM), which resulted in a quick and timely precise receptor activation (Figure 5D). Moreover, upon activation, HGFR underwent degradation, becoming undetectable 90 min after HGF addition. Then, we performed time-of-addition experiments, adding 50 nM HGF before, during, and after rLCMV/LASVGP infection. The addition of HGF before or after viral infection did not affected rLCMV/LASV_GP_ relative infectivity, whereas its presence during viral entry increased rLCMV/LASV_GP_ entry in approximately 40% (Figure 5E and Appendix A). In contrast, infectivity of rLCMV-VSV_G_, which enters cells independently of HGFR [27], was not affected by the presence of HGF at any of the time points analyzed (Figure 5E and Appendix A).

Next, we sought to investigate if HGFR directly interacted with viral particles during infection. To that aim, precooled (4 °C) A549 cells were infected with rLCMV/LASV_GP_, for 2 h, at 4 °C, and shifted to 37 °C for indicated times. Then, cells were fixed at 0, 2, 5, or 10 min after infection, and stained with specific antibodies for HGFR and viral NP detection. Similar to previous visualization of Axl RTK and NP, HGFR did not increase colocalization with NP over time, after infection (Figure 5F). Moreover, we followed up HGFR/DG clustering upon viral infection in A549 cells and found that HGFR and DG did not increase their colocalization due to rLCMV/LASV_GP_ infection (Figure 5G).

To further assess the relative contributions of Axl and HGFR RTKs during LASV entry, in the absence and presence of functional DG, and a potential interaction between both receptors, we performed infections with rLCMV/LASV_GP_ in the presence of EMD (10 µM) and increasing concentrations of R428. The results showed that regardless of the presence or absence of functional DG EMD significantly enhanced the antiviral effect of R428 (Figure 5H). The antiviral effect of either R428 alone or in combination with 10 µM of EMD was more pronounced in HT1080 than in A549 cells. To determine the nature of the interaction, antagonistic, additive or synergistic, between both RTK inhibitors, we analyzed out results with CompuSyn software [95]. The analysis revealed that in both cell lines, A549 and HT-1080 cells, R428, and EMD had an additive effect one to the other (CI values approximately 1) (Table 1 and Table 2).

LASV enters cells through a macropinocytosis-related pathway [27,37]. Thus, given the evidence of involvement of Axl and HGFR RTKs in LASV entry, next, we addressed the relative contribution of both RTKs to classical macropinocytosis. In most cells, macropinocytosis occurs at basal levels and can be tuned through different stimuli, including activation of specific RTKs by growth factors [96,97,98]. RTK signaling results in activation of Rho GTPases and PAK1, leading to actin rearrangement and macropinocytotic cup formation [98]. To assess the role of Axl and HGFR RTKs in macropinocytosis, we treated A549 cells with R428 or EMD, respectively, and monitored macropinosome formation and trafficking with the fluid marker Alexa Fluor 488-conjugated dextran [99]. In agreement with its mechanism of action, EIPA significantly reduced the number of dextran-labeled vesicles [100] (Figure 6A).

Next, we assessed the effect of EIPA, R428, and EMD on rLCMV/LASV_GP_ cellular internalization. To that aim, A549 cells were incubated in the cold with the virus at high MOI (50 PFU/cell), and the temperature was subsequently shifted to 37 °C, for 40 min, to allow virus internalization in the presence of the aforementioned drugs. Then, viral NP was stained with a specific antibody. The RTK inhibitors, R428 and EMD, as well as EIPA, all prevented virus internalization, resulting in accumulation of viral particles at proximity of the cellular membrane (Figure 6B). Despite the different pattern in dextran-containing vesicles observed among R428, EMD, and EIPA, the three drugs lead to comparable patterns, with accumulation of virion-containing vesicles at the membrane proximity (Figure 6A,B). In summary, these results reveal that Axl and HGFR RTKs participate in virus internalization through macropinocytosis, and their inhibition consequently leads to inhibition of trafficking of virus-containing vesicles through the cytoplasm.

### 3.4. R428 and EMD Represent Promising Antiviral Candidates against Lassa Fever

Our results, so far, revealed a cooperative inhibitory effect of EMD and R428 against LASV entry in the presence or absence of functional DG, by affecting macropinosome trafficking. Antiviral cocktails to combat RNA virus infection are especially powerful, when cocktails contain combinations of drugs targeting different steps of the viral cycle [101,102,103]. To date, the off-label administration of Rib is the only available treatment against LASV infections [4]. To investigate the antiviral activity of RTKs in combination with Rib, we pretreated A549 and HT-1080 cells with indicated drugs for 30 min at 37 °C. Upon drug preincubation, to synchronize infections, cells were quickly cooled down to 4 °C and infected at 4 °C with rLCMV/LASV_GP_ in the presence of increasing concentrations of Rib (0–20 µM), either alone (DMSO) or in combination with EMD (5 µM) and increasing concentrations of R428 (0–1 µM), and shifted to 37 °C, for 1 h. After infection, cells were incubated for 16–20 h at 37 °C in the presence of indicated drug concentrations and 20 mM of ammonium chloride. The results showed that R428 and EMD increased the antiviral activity of low to mild doses of Rib in both cell lines in the absence of overall toxicity (Figure 7 and Appendix A). 

In agreement with previous results, the effect was more pronounced in HT-1080 than in A549 cells. To further determine the type of interaction, antagonistic, additive or synergistic, among the drugs present in the cocktail, we performed further analysis with CompuSyn [95]. In A549 cells, all R428/EMD and Rib combinations rendered CI values < 1, suggesting synergistic interaction between the RTK inhibitors and Rib (Table 3).

In HT-1080 cells, moderate R428/EMD cocktail concentrations (R428 0.125-1 µM + EMD 5 µM) interacted synergistically with Rib, whereas low and high cocktail concentrations were closer to the additive effect (CI values approximately 1, Table 4).

## 4. Discussion

The ECM receptor DG is the main receptor for the OW mammarenaviruses LASV and LCMV [14,15] and virus binding occurs through the matriglycan chains exposed on the cellular membrane [24]. Although expressed in most cell types, DG is subjected to complex, tightly regulated post-translational glycosylation modifications, including extensive O-mannosyl-linked carbohydrate addition [104], which critically determines DG functionality [19]. High affinity virus-receptor binding occurs rapidly via long matriglycan chains linked to the N-terminus of the mature DG protein, and virus-receptor dissociation occurs only under low pH, pointing to internalization of the virus-DG complex [29,30,31]. Nevertheless, identified infected tissues do not strictly correlate with expression levels of functional DG [7,33], suggesting that LASV entry does not depend only on DG but also on additional cellular factors, which can act in a coordinated manner depending on the specific tissue. Indeed, Axl and HGFR RTKs, which are expressed in many target cells of LASV, have been identified as additional host factors important for viral entry [27,36,37]. In the present study, we investigated the function of these RTKs in productive entry of LASV in the context of presence or absence of functional glycosylated DG.

Axl RTK participated in the entry of several emerging viruses, such as Ebola, Zika, and dengue [50,51,52,105,106], and we have previously shown that Axl RTK contributed to LASV entry in cells lacking functional DG [37]. In the present study, we have provided evidence that, regardless of the DG glycosylation extent, Axl RTK is required for LASV entry but not for later steps of viral replication, and that Axl RTK inhibition consequently resulted in decreased rLCMV/LASV_GP_ infectivity. Indeed, in HT-1080 cells, which do not express functional glycosylated DG, the inhibition of Axl RTK by R428 resulted in a stronger antiviral effect than in A549 and HUVEC cells, which displayed glycosylated forms of DG. Confirming our initial observations, we found that Axl RTK overexpression resulted in increased rLCMV/LASV_GP_ infectivity, whereas in a complementary approach, Axl RTK knockdown reduced viral infectivity. Nevertheless, although we observed rLCMV-LASV_GP_ colocalization with preformed DG clusters at the plasma membrane, virions did not seem to increase colocalization over time with Axl RTK. This is in agreement with previous observations of LASV infectivity reduction due to effective antibody perturbation targeting DG but not Axl RTK [80] and suggests, at best, transient, weak interactions between LASV_GP_ and Axl RTK. Nevertheless, it could be conceivable that LASV induces DG clustering, followed by the recruitment of additional entry factors or co-receptors, for example, Axl RTK, to the initial site of attachment. Therefore, we assessed if LASV membrane attachment and subsequent internalization would increase DG-Axl colocalization, which was not the case. Nevertheless, it could be that the basal activity of Axl RTK is required for LASV virion trafficking through the cytoplasm, and thus its inhibition abolishes virion-containing vesicles transportation. This would be in agreement with the lack of colocalization of viral particles or DG with Axl RTK, as well as with all results that demonstrate Axl RTK participation in LASV entry. Moreover, as R428 inhibited R428 catalytic activity [83], it is suggested that this activity is specifically required for LASV entry.

Previous and present findings have indicated that LASV entered target cells by a macropinocytosis-related pathway [27,80] and that the inhibition of HGFR, a RTK involved in macropinocytosis [64], resulted in LASV entry reduction regardless the extent of DG glycosylation [27]. The reduced macropinosome trafficking induced by inhibition of Axl or HGFR RTKs suggested that R428 and EMD exerted their antiviral effects by interfering with cytoplasmic transport of virus-containing vesicles, after initial cellular attachment and internalization of LASV. The observed basal macropinosome and LASV virion-containing vesicle traffic observed in untreated cells, and its inhibition by R428 and EMD, further supported the hypothesis that Axl and HGFR RTKs basal signaling was required for LASV transportation. Furthermore, we found that both RTK inhibitors, EMD and R428, displayed modest additive antiviral effects, resulting in strong inhibition of rLCMV/LASV_GP_ entry. LASV infection is associated with liver function impairment [8,9]; therefore, although EMD is already in clinical trials for cancer therapy [84,86], administration of EMD for LHF treatment should be finely controlled to minimize potential side effects on liver physiology. Nevertheless, the possible combinations with other antivirals could allow one to reduce EMD doses to those that are able to reduce LASV infection while minimizing the undesired effects at the hepatic function. Indeed, cocktail treatments with both RTK inhibitors and Rib revealed synergistic interactions, allowing reduction of drug doses and suggesting potential for further development of antiviral therapies.

A previous study [35] found that Tim-1 and Axl RTK did not contribute to LASV pseudovirus transduction of 293T cells expressing highly glycosylated DG, which represented an apparent contradiction with our studies. The Axl RTK and Tim-1 which both belong to the TAM and TIM receptor families, respectively, thus, bind PS and can support viral entry by binding to the viral phospholipid membrane, depending on the relative GP/PS abundance on the virion surface and the virion architecture. Pseudoviruses are powerful tools in virology in which a virus lacking its envelope glycoprotein is used as a platform to generate pseudoviruses decorated with heterologous glycoproteins. Therefore, many factors affect pseudoviral production, including the expression level of ectopic GP, or the phospholipid composition of viral membranes, which can affect virus behavior, phenotype, and receptor usage. To prevent artificial virus–cell interactions, in our study, we used a replication-competent rLCMV/LASV_GP_ chimera, which incorporated LASV_GP_ in the backbone of the close relative OW mammarenavirus LCMV and faithfully recapitulated LASV infection in vitro and in vivo [66,77]. Alternatively, it may be possible that exogenously expressed Axl RTK does not completely recapitulate the cellular factor functionality, which provides a possible explanation for both findings.

Axl RTK activation is associated with reduction of the IFN-I response, resulting in increased viral replication of many viruses [51,93,94]. Nevertheless, our results show that, in our experimental settings, Axl RTK inhibition by R428 inhibits rLCMV/LASV_GP_ entry without affecting IFN-I expression. It is conceivable that the well-known immunosuppressive capacity of LASV (REF) is potent enough to also prevent the effect of R428 on the innate immune response. Nevertheless, at late phases of the infection, the host can detect LASV presence, triggering a strong innate immune response [107]. Therefore, the inhibition of Axl RTK could play a relevant role at late infection times. Our time-of-addition and post-entry assays, as well as the virus visualization under confocal microscopy, instead, suggest that Axl RTK participates mainly during LASV transport, early after virion attachment.

In summary, our study provides evidence for the participation of Axl and HGFR RTKs in LASV entry, but not in later steps, regardless of the presence or absence of functional, highly glycosylated DG. Furthermore, inhibition of both RTKs, Axl, and HGFR, resulted in reduced LASV entry by alteration of cytoplasmic macropinosome trafficking with vesicle retention and accumulation at the cellular membrane. The present results suggest that basal macropinocytosis, which involve Axl and HGFR RTKs, is necessary for LASV transportation upon viral internalization, and highlights the importance of additional host factors in the process of viral entry and their relevance as promising antiviral drug targets.

## Figures and Tables

**Figure 1 viruses-12-00857-f001:**
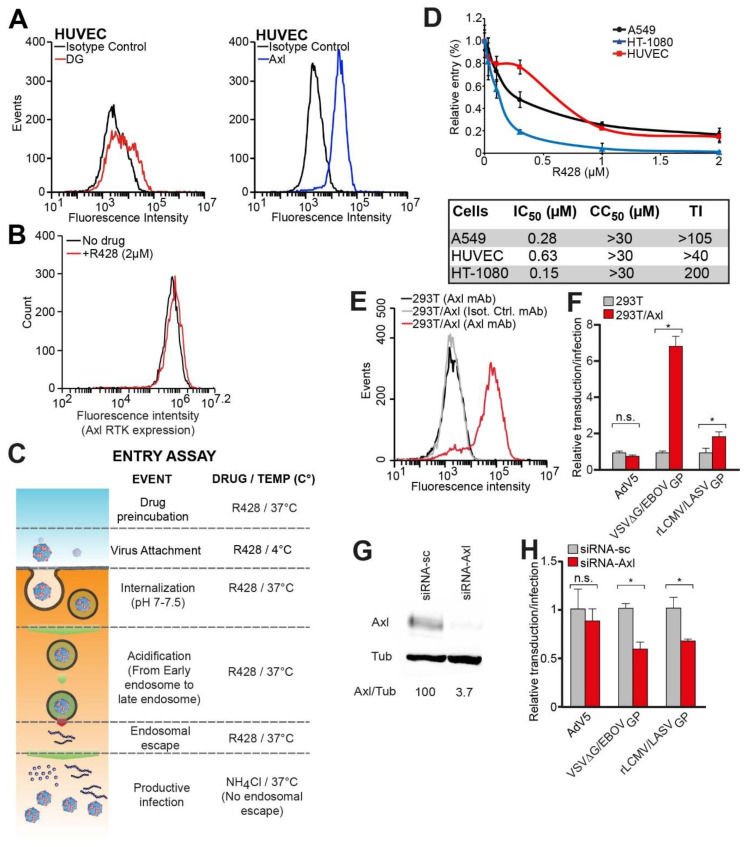
Axl receptor tyrosine kinase (RTK) participates in LASV infection. (**A**) Surface expression of glycosylated dystroglycan (DG) and Axl RTK in HUVEC cells. Live, nonpermeabilized cells were stained at 4 °C with specific antibodies to sugar chains of DG or Axl RTK and analyzed by fluorescence-activated cell sorting (FACS); (**B**) Surface expression of Axl RTK in A549 cells treated with R428. Cells were incubated with 2 µM of R428 for 1.5 h, and then fixed and stained with the specific anti-Axl RTK antibody; (**C**) Schematic experimental setting of the entry assay; (**D**) Dose-response curves in entry assays for R428 inhibition of rLCMV/LASV_GP_ infection in A549, HT-1080, and HUVEC cells. Error bars represent standard deviations (*n* = 3); (**E**) FACS analysis of Axl RTK surface expression in parental and 293T with Axl-constitutive expression; (**F**) Relative entry of vesicular stomatitis virus (VSV)-based pseudotype decorated with EBOV_GP_ (VSVΔG/EBOV_GP_), rLCMV/LASV_GP_, and of AdV5 in Axl-transfected 293T cells. Error bars represent standard deviations (*n* = 3). Asterisk (*) denotes *p*-value < 0.01 (Student’s *t*-test); (**G**) Depletion of Axl RTK from A549 cells. Two days post siRNA transfection, expression of Axl RTK was detected by Western blot, using α-tubulin (Tub) as the loading control. Efficiency of Axl RTK depletion was assessed by densitometry using Image J. Numbers beneath the blot indicate the percentages of the signal ratio of Axl/α-tubulin; (**H**) Infection of Axl-siRNA-depleted A549 cells. Cells transfected with siRNA-Axl or siRNA-sc (scramble, negative control) were infected with AdV5-EGFP, VSVΔG-EBOV_GP_, or rLCMV/LASV_GP_. Error bars represent standard deviations (*n* = 3). Asterisk (*) denotes *p*-value < 0.01 (Student’s *t*-test).

**Figure 2 viruses-12-00857-f002:**
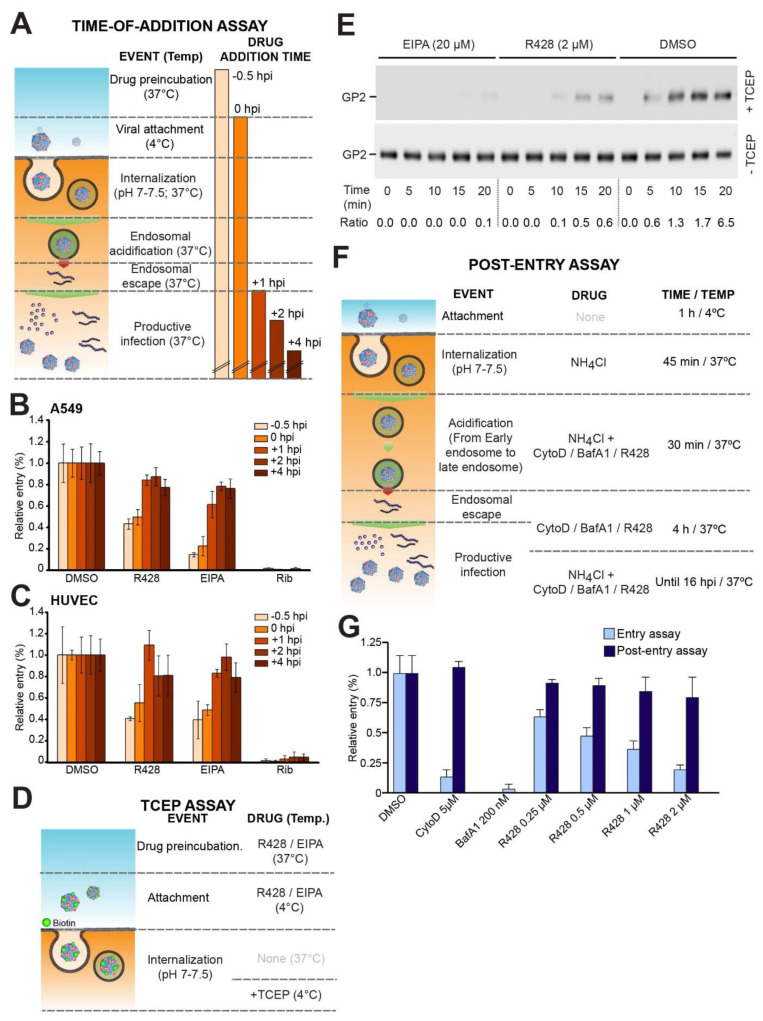
Axl RTK participates only in early steps of rLCMV/LASV_GP_ entry. (**A**) Schematic experimental setting of the time-of-addition assay. R428, 5-(*N*-ethyl-*N*-isopropyl)-amiloride (EIPA), or ribavirin (Rib) were added at the indicated times relative to infection. Ammonium chloride (20 mM) was added 1 h after infection (+1 hpi) to prevent reinfections; (**B**,**C**) Inhibition of rLCMV/LASV_GP_ infection in time-of-addition assays by R428 (0.5 µM), EIPA (20 µM), and Rib (100 µM) in A549 (**B**), and HUVEC cells (**C**); (**D**) Schematic experimental setting of the TCEP assay to monitor virus internalization, (**E**) Viral uptake in the presence of EIPA and R428 in A549 cells. After quenching of residual TCEP, cells were lysed, and viral GP isolated by IP with mAb 83.6 to LCMV GP2. Biotinylated GP2 was detected with streptavidine-HRP in Western blot under non-reducing conditions using enhanced chemiluminescence (ECL). The upper blot (+TCEP) was exposed for 5 min and the lower blot (−TCEP) for 30 s. Ratios of +TCEP/−TCEP signals (densitometry) were calculated using ImageJ software; (**F**) Schematic experimental setting of the post-entry assay; (**G**) Relative entry of rLCMV/LASV_GP_ in entry and post-entry assays in the presence of CytoD, BafA1, or R428. Values were normalized with those obtained in the presence of dimethyl sulfoxide (DMSO). Error bars represent standard deviations (*n* = 3).

**Figure 3 viruses-12-00857-f003:**
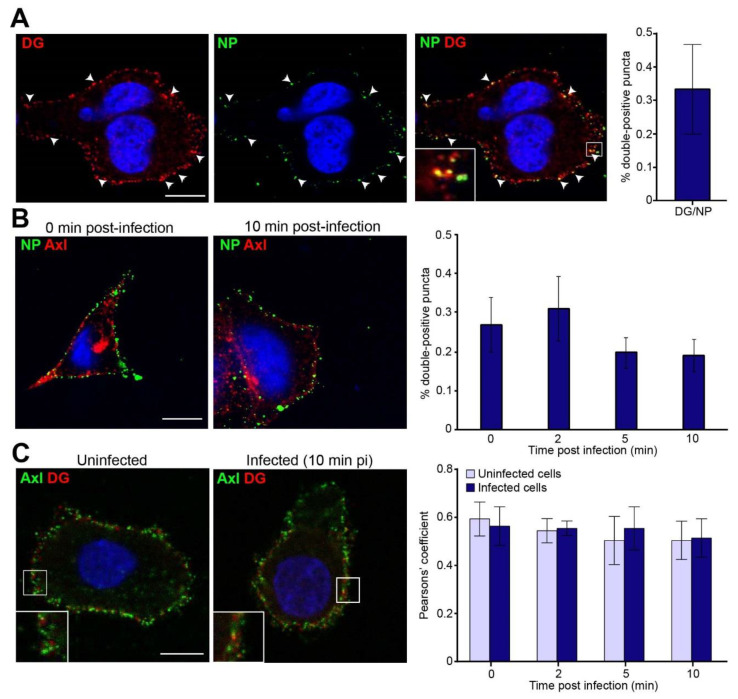
Colocalization of rLCMV/LASV_GP_ viral particles, DG, and Axl RTK. (**A**) DG/nucleoprotein (NP) and (**B**) Axl RTK/NP colocalization. Precooled (4 °C) A549 cells were infected, for 2 h, in the cold with rLCMV/LASV_GP_ (MOI 50 PFU/cell), shifted to 37 °C for indicated times, and fixed. DG (A) or Axl RTK (B) were visualized in red and viral NP in green (**A**,**B**). Representative confocal images are shown. Scale bar represents 5 µm. Double puncta percentages were obtained from the analysis of 20 (**A**) and 8 (**B**) randomly selected cells per condition. Error bars represent standard error; (**C**) DG/Axl RTK colocalization in mock-infected and rLCMV/LASV_GP_-infected cells. Live A549 cells were preincubated, for 2 h, in the cold with mock or rLCMV/LASV_GP_ (MOI 50 PFU/cell), and rapidly shifted to 37 °C for indicated times. DG is visualized in red and Axl RTK in green. Representative images collected by confocal microscopy are shown. Scale bar represents 5 µm. Colocalization of DG and Axl RTK was quantified by Pearson’s coefficient in mock and infected cells after shifting to 37 °C, for 0, 2, 5, and 10 min. Error bars represent standard deviations of 10 randomly selected cells per condition. Axl does not contribute to productive LASV entry via Interferon-I suppression.

**Figure 4 viruses-12-00857-f004:**
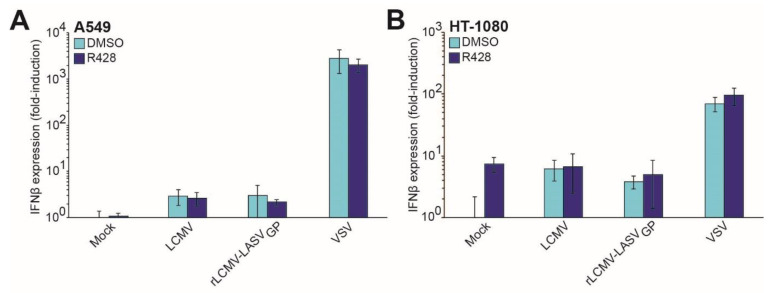
Axl RTK does not affect the interferon-I response. (**A**) A549 or (**B**) HT-1080 cells were pretreated with DMSO or 2 μM R428, 30 min before viral infection, and rLCMV/LASV_GP_ (MOI 0.01 PFU/cell), LCMV Arm 53b (MOI 0.01 PFU/cell), or VSV (MOI 1 PFU/cell) were added in presence of DMSO or 2 μM R428. After infection, 20 mM ammonium chloride were added to the media to prevent multiple rounds of replication. Total RNA was extracted 16 h after infection, reverse transcribed, and qPCR performed to determine interferon β (INF β) expression levels. Error bars represent standard deviations (*n* = 6). Axl and hepatocyte growth factor receptor cooperate during LASV entry.

**Figure 5 viruses-12-00857-f005:**
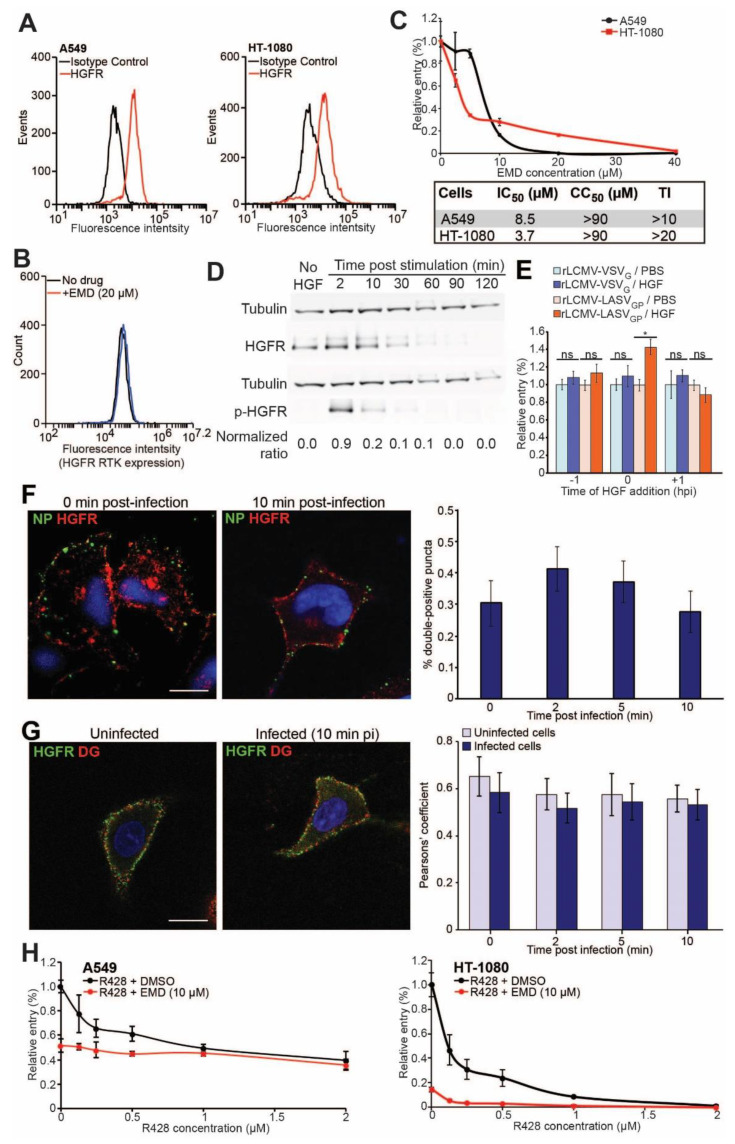
Axl and hepatocyte growth factor receptor (HGFR) RTKs cooperate during LASV entry. (**A**) HGFR surface expression in A549 and HT-1080 cells detected by FACS analysis; (**B**) HGFR surface expression upon EMD1204063 (EMD) administration. A549 cells were incubated for 1.5 h with EMD (20 µM), and then fixed and stained for HGFR detection with specific antibody; (**C**) Dose dependent entry inhibition of rLCMV/LASV_GP_ by EMD in A549 and HT-1080 cells. Error bars represent standard deviations (*n* = 3); (**D**) Phosphorylation and expression of HGFR after HGF administration. Cells were incubated with HGFR for indicated times, lysed, and tested by WB for HGFR and phosphorylated HGFR (p-HGFR). Tubulin was used as loading control. Normalized ratios were calculated from signal ratios of (HGFR/tubulin)/(p-HGFR/tubulin); (**E**) Time-of-addition assay of HGF. HGF (50 nM) or vehicle (PBS) were added to A549 cells at the indicated times relative to infection with rLCMV-VSV_G_ or rLCMV/LASV_GP_. Error bars represent standard deviations (*n* = 3). Asterisk (*) denotes significant differences (student’s *t*-test; *p* < 0.01); (**F**) HGFR RTK/NP colocalization. Precooled (4 °C) A549 cells were infected, for 2 h, in the cold with rLCMV/LASV_GP_ (MOI 50 PFU/cell), and then shifted to 37 °C for indicated times. HGFR RTK was visualized in red and viral NP in green. Representative confocal images are shown. Scale bar represents 5 µm. Percentages of double puncta (NP/HGFR RTK) values obtained from the analysis of 8 randomly selected cells per condition. Error bars represent standard deviations; (**G**) DG/HGFR RTK colocalization in mock-infected and rLCMV/LASV_GP_-infected cells. Live A549 cells were preincubated for 2 h in the cold with mock or rLCMV/LASV_GP_ (MOI 50 PFU/cell), and rapidly shifted to 37 °C during indicated times. DG is visualized in red and Axl RTK in green. Representative images collected by confocal microscopy are shown. Scale bar represents 5 µm. Colocalization of DG and HGFR RTK was quantified by Pearson’s coefficient in mock and infected cells after shifting to 37 °C, for 0, 2, 5, and 10 min. Error bars represent standard deviations of 10 randomly selected cells per condition. Scale bars represent 5 µm; (**H**) Dose response inhibition of rLCMV/LASV_GP_ relative entry into A549 and HT-1080 cells by R428, alone or in combination with EMD (10 µM). Relative entry values were normalized to the infectivity in the absence of drug (DMSO). Error bars represent standard deviations (*n* = 3).

**Figure 6 viruses-12-00857-f006:**
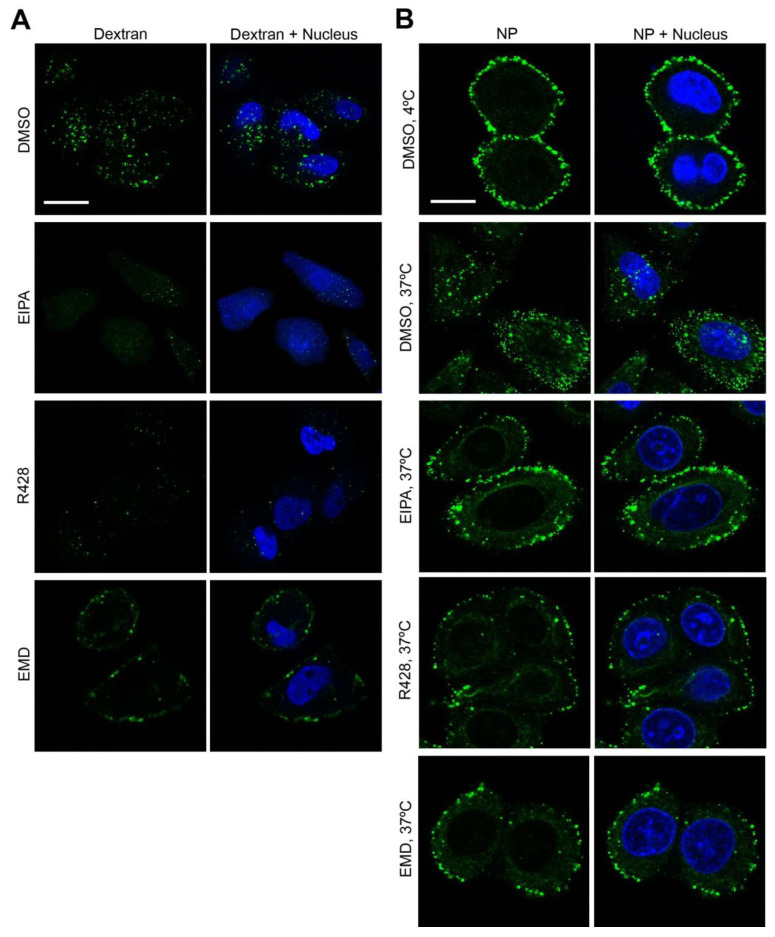
R428 and EMD inhibit macropinocytosis-related rLCMV/LASV_GP_ internalization. (**A**) A549 cells were pretreated with DMSO, 40 µM EIPA, 5 µM R428, or 40 µM EMD, for 30 min. Subsequently, cells were incubated with Alexa Fluor 488 labeled dextran (250 µg/mL) in the presence of the inhibitors, for 40 min, at 37 °C, washed, fixed with PFA, and subjected to confocal microscopy. Note: EIPA treatment resulted in a faint blue fluorescence *throughout* the cytosol. Representative fields for each condition are shown. Scale bar: 10 µm; (**B**) Precooled A549 cells were incubated for 2 h, at 4 °C, with rLCMV/LASV_GP_ (MOI 50 PFU/cell). Cells were washed and either fixed or shifted to 37 °C, for 45 min, in the presence of DMSO or inhibitors (same concentrations as in (**A**)), as indicated. Then, cells were fixed, stained for viral NP (113 antibody), and analyzed by confocal microscopy. Representative fields are shown for each condition (*n* = 10). Scale bar corresponds to 10 µm. Interestingly, whereas Axl RTK inhibition by R428 decreased the number of dextran-labeled vesicles, HGFR inhibition by EMD resulted in an accumulation of vesicles in proximity to the plasma membrane (Figure 6A). In agreement with previous reports [27,37], we also found in viral infection experiments that 18.6% (±7.7%) of total Alexa Fluor 488-conjugated, dextran-containing vesicles also contained rLCMV/LASV_GP_ viral particles (Appendix A).

**Figure 7 viruses-12-00857-f007:**
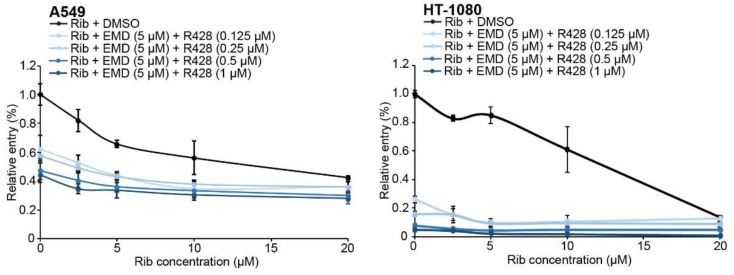
Antiviral effect of R428, EMD, and Rib cocktails. A549 and HT-1080 cells were pretreated for 30 min, at 37 °C, cooled down to 4 °C, and infected with rLCMV/LASV_GP_ in the presence of indicated concentrations of Rib, either in combination with DMSO or EMD (10 µM) and R428 (0–1 µM). Then, cells were moved to 37 °C, for 1 h. Upon infection, cells were incubated for 16–20 h in the presence of indicated drugs and 20 mM of ammonium chloride. Relative entry values were normalized to the infectivity in absence of any drug. Error bars represent standard deviations (*n* = 3).

**Table 1 viruses-12-00857-t001:** Drug interaction (R428/EMD) analysis in A549 cells. Analysis performed with CompuSyn [95]. CI values <1 means synergistic interaction, approximately 1 additive effect, and >1 antagonistic interaction.

A549 Cells			
R428 Conc. (µM)	EMD Conc. (µM)	Effect	CI Value
0.125	10.0	0.49	1.20910
0.25	10.0	0.52	1.24309
0.5	10.0	0.55	1.33485
1.0	10.0	0.54	1.78463
2.0	10.0	0.64	1.59548

**Table 2 viruses-12-00857-t002:** Drug interaction (R428/EMD) analysis in HT-1080 cells. Analysis performed with CompuSyn [95]. CI values <1 means synergistic interaction, approximately 1 additive effect, and >1 antagonistic interactionAxl and HGFR participate in classical macropinocytosis.

HT-1080 Cells			
R428 Conc. (µM)	EMD Conc. (µM)	Effect	CI Value
0.125	10.0	0.95	1.20408
0.25	10.0	0.97	1.12084
0.5	10.0	0.97	1.28517
1.0	10.0	0.99	1.04369
2.0	10.0	0.999	0.554

**Table 3 viruses-12-00857-t003:** Drug interaction (R428 + EMD/Rib) analysis in A549 cells. Analysis performed with CompuSyn [95]. CI values < 1 means synergistic interaction, approximately 1 additive effect, and >1 antagonistic interaction.

A549 Cells			
Rib Concentration (µM)	R428 + EMD (5 µM) Concentration (µM)	Effect	CI Value
2.5	0.125	0.47	0.14113
2.5	0.25	0.51	0.23307
2.5	0.5	0.6	0.31581
2.5	1.0	0.64	0.52867
5.0	0.125	0.56	0.09568
5.0	0.25	0.57	0.18109
5.0	0.5	0.64	0.26473
5.0	1.0	0.67	0.46052
10.0	0.125	0.65	0.06400
10.0	0.25	0.62	0.14596
10.0	0.5	0.67	0.23072
10.0	1.0	0.7	0.39856
20.0	0.125	0.64	0.06818
20.0	0.25	0.64	0.13423
20.0	0.5	0.7	0.19979
20.0	1.0	0.72	0.36039

**Table 4 viruses-12-00857-t004:** Drug interaction (R428 + EMD/Rib) analysis in HT-1080 cells. Analysis performed with CompuSyn [95]. CI values < 1 means synergistic interaction, approximately 1 additive effect, and >1 antagonistic interaction.

HT1080			
Rib Concentration (µM)	R428 + EMD (5 µM) Concentration (µM)	Effect	CI Value
2.5	0.125	0.85	0.61251
2.5	0.25	0.85	1.12848
2.5	0.5	0.95	0.66368
2.5	1.0	0.96	1.01803
5.0	0.125	0.91	0.42575
5.0	0.25	0.91	0.71527
5.0	0.5	0.96	0.56971
5.0	1.0	0.98	0.53217
10.0	0.125	0.9	0.61743
10.0	0.25	0.91	0.85150
10.0	0.5	0.96	0.65063
10.0	1.0	0.99	0.30661
20.0	0.125	0.88	1.06018
20.0	0.25	0.92	1.01350
20.0	0.5	0.96	0.81248
20.0	1.0	0.99	0.37553

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
