# Peer review of "The Role of Receptor Tyrosine Kinases in Lassa Virus Cell Entry"

_viruses, 2020, doi:10.3390/v12080857_

Round 1
Reviewer 1 Report
This manuscript from Fedeli et al studies relative contributions of the receptor tyrosine kinases (RTK) Axl and HGFR towards Lassa virus (LASV) cell entry in comparison to dystroglycan (DG). Lassa virus continues to cause cases of severe disease in the absence of an approved vaccine. The only available treatment for Lassa fever is the general anti-viral drug Ribavirin, which can cause significant side effects and is effective only when given in early and non-specific stages of disease. Novel antivirals, with a particular focus on anti-viral drugs that can block arenaviral cell entry to avoid Lassa virus- associated immunosuppression and pathogenesis, are therefore a critical topic of research on Lassa virus.
Studies have consistently found that the primary cellular receptor for Lassa virus entry is the extra-cellular matrix protein dystroglycan (DG). However, it has also been found that DG may be under several regulatory pathways that further determine susceptibility of Lassa virus entry. In addition to dynamic levels of DG glycosylation, preliminary evidence suggests that additional cellular proteins such as the TAM and TIM families of phosphatidylserine receptors may also play a role in Lassa virus cell entry. In particular, the TAM receptor tyrosine kinase (TRK) Axl has been found to contribute to Lassa virus cell entry particularly in the context of un-glycosylated DG using a macropinocytosis-like pathway that may be similar to that seen with entry through DG. Additionally, the RTK HGFR has been found to participate in macropinocytosis and has previously exhibited antiviral effects against Lassa virus. However, their specific contributions towards Lassa virus entry in relation to DG remains unknown.
This manuscript uses a number of methods to study the mechanisms of Lassa virus entry mediated by Axl and HGFR and their contribution relative to DG. Their conclusion is that both factors can allow Lassa virus entry in the absence of functional DG, but that all three factors play an additive role to Lassa virus entry. As a whole, their experimental flow-through is straightforward, and the background information to justify their experiments are thoroughly explained. However, there are some points related to experimental design and reporting of data that should be addressed prior to acceptation for publication.
Major points
- In figures 1B, 2A, 2E, 5B, 5D and 7 the authors didn’t specify whether they incubated the cells on ice following infection (which they did note for the experiments in figures 2G, 3 and 6), which is necessary to ensure consistency across their experimental conditions by controlling the timing of cell entry.
- On page 4, the authors don’t specify whether expressing mutant forms of Axl was also cytotoxic in 293T cells- if not, this would be an alternative loss of function assay to performing it in the more biologically relevant cell lines.
- A number of figures only report relative values when raw values are also needed to confirm that the data lies within the functional range of the assay- these figures include 1C, 1E, 1G, 2B-D, 2H, 4A-B, 5B-D and 7
- In figures 2B-D, the use of ribavirin doesn’t appear to act as an appropriate control of post-entry steps for R428 and EIPA since ribavirin inhibited entry at all time points (instead of only at later timepoints) and it inhibited Lassa entry at significantly higher levels than R428 and EIPA. An alternative drug may need to be used as a better control, or ribavirin may need to be given at later time points or in smaller amounts to counteract longer-lasting effects
- In figure 2E, the authors should justify using the TCEP biotin assay to assess virus internalization over imaging or determining viral titer after leaving the cells on ice, which would be more quantifiable. In fact in reference 69, the authors using this same assay specify that they left the cells on ice both during incubation with the virus and with incubation with TCEP, which allowed for a timed visualization of cell entry. This timed visualization isn’t possible in 2F since biotin signal from internalized virus could be seen in the DMSO treatment condition 0 minutes after TCEP quenching.
- In figures 3A and B, Pearson’s coefficient may not be the best assessment of NP colocalization since the DG and Axl signals are significantly greater and more diffuse than the NP signal. This is reflected in the moderate difference in Pearson’s coefficient between 3A and 3B. A better assessment may be NP-DG or Axl double-positive puncta, which appears to be the biggest difference between 3A and 3B.
- In figure 5D, the authors could include a control to indicate that EMD truly has an additive effect specific with R428 as entry inhibitors as compared to any anti-Lassa drug
- In figure 7, it seems like there may be more use in varying the concentration of R248 at more points compared to ribavirin since it has a lower concentration of use and therefore might be more viable as a drug candidate. It would also be more likely to generate differences than two points at a 2x dilution.
Minor points
- On page 2 line 93, the authors should explain in more detail the concept of “apoptotic mimicry” and how that relates to viral entry.
- On page 3 line 101, the authors should briefly explain the context that Axl was thought to co-operate with DG and how under-glycosylation of DG affected the interaction of Axl and DG.
- On page 4 line 159, the authors should provide a citation if Mer/Tyro3 upregulation has been seen following Axl inhibition.
- On page 4 line 171, the authors should explain why an Ebola virus construct was necessary to infect 293T cells.
- On page 8 line 246, it can’t be determined whether DG clusters are truly “preformed” prior to interaction with LCMV/LASV- the word should be removed.
- On page 10 line 279, it should also be noted that R248 appeared to induce an IFNB expression in mock infected HT-1080 cells, possibly making R248 in HT-1080 cells more complicated to study viral replication than in other cell types
- On page 10 line 307, it may be misleading to say that HGF significantly promoted infection with a 40% increase. The authors should also address why the timing of HGF appeared to be more specific than the drug treatments used throughout the manuscript.
- On page 13 line 351, EIPA and R248 also seemed to have NP puncta in proximity to the plasma membrane- the authors should explain any differences they noted with EMD compared to these two drugs in this regard.
- On page 14 line 410, the “land and stick” usage of words didn’t make sense for the model being described.
- On page 15 line 433, the results in this manuscript aren’t necessarily in contradiction with earlier results since the earlier results were performed in 293T cells expressing highly glycosylated DG- DG might take precedence over Axl and HGFR.
Author Response
Dear Sir, dear Madam,
Please, find below the addressed comments from Reviewers’ revision of the manuscript entitled « ROLE OF RECEPTOR TYROSINE KINASES IN LASSA VIRUS CELL ENTRY".
We appreciate very much the critical and constructive comments from all Reviewers. In addition to the indicated modifications to specifically address the Reviewer’s comments, we have also included other changes in the manuscript to make a clearer description of our findings.
Sincerely yours,
Héctor Moreno, PhD.
Lausanne, 31.7.2020.
REVIEWER 1
Open Review
(x) I would not like to sign my review report
( ) I would like to sign my review report
English language and style
( ) Extensive editing of English language and style required
( ) Moderate English changes required
(x) English language and style are fine/minor spell check required
( ) I don't feel qualified to judge about the English language and style
|
Yes |
Can be improved |
Must be improved |
Not applicable |
|
|
Does the introduction provide sufficient background and include all relevant references? |
(x) |
( ) |
( ) |
( ) |
|
Is the research design appropriate? |
( ) |
(x) |
( ) |
( ) |
|
Are the methods adequately described? |
( ) |
( ) |
(x) |
( ) |
|
Are the results clearly presented? |
( ) |
(x) |
( ) |
( ) |
|
Are the conclusions supported by the results? |
( ) |
(x) |
( ) |
( ) |
Comments and Suggestions for Authors
This manuscript from Fedeli et al studies relative contributions of the receptor tyrosine kinases (RTK) Axl and HGFR towards Lassa virus (LASV) cell entry in comparison to dystroglycan (DG). Lassa virus continues to cause cases of severe disease in the absence of an approved vaccine. The only available treatment for Lassa fever is the general anti-viral drug Ribavirin, which can cause significant side effects and is effective only when given in early and non-specific stages of disease. Novel antivirals, with a particular focus on anti-viral drugs that can block arenaviral cell entry to avoid Lassa virus- associated immunosuppression and pathogenesis, are therefore a critical topic of research on Lassa virus.
Studies have consistently found that the primary cellular receptor for Lassa virus entry is the extra-cellular matrix protein dystroglycan (DG). However, it has also been found that DG may be under several regulatory pathways that further determine susceptibility of Lassa virus entry. In addition to dynamic levels of DG glycosylation, preliminary evidence suggests that additional cellular proteins such as the TAM and TIM families of phosphatidylserine receptors may also play a role in Lassa virus cell entry. In particular, the TAM receptor tyrosine kinase (TRK) Axl has been found to contribute to Lassa virus cell entry particularly in the context of un-glycosylated DG using a macropinocytosis-like pathway that may be similar to that seen with entry through DG. Additionally, the RTK HGFR has been found to participate in macropinocytosis and has previously exhibited antiviral effects against Lassa virus. However, their specific contributions towards Lassa virus entry in relation to DG remains unknown.
This manuscript uses a number of methods to study the mechanisms of Lassa virus entry mediated by Axl and HGFR and their contribution relative to DG. Their conclusion is that both factors can allow Lassa virus entry in the absence of functional DG, but that all three factors play an additive role to Lassa virus entry. As a whole, their experimental flow-through is straightforward, and the background information to justify their experiments are thoroughly explained. However, there are some points related to experimental design and reporting of data that should be addressed prior to acceptation for publication.
Major points
- Question / Comment: In figures 1B, 2A, 2E, 5B, 5D and 7 the authors didn’t specify whether they incubated the cells on ice following infection (which they did note for the experiments in figures 2G, 3 and 6), which is necessary to ensure consistency across their experimental conditions by controlling the timing of cell entry.
Response: Following the Reviewer’s comment, and to clarify the protocols in the manuscript, mentioned figures are now modified to include the temperature of each step, and detailed descriptions of “Entry”, “post-entry”, “time of addition” and “TCEP” assays are included in lines 164-71, 210-17, 240-47, 328-31, 371-4 and 601-618.
- Question / Comment: On page 4, the authors don’t specify whether expressing mutant forms of Axl was also cytotoxic in 293T cells- if not, this would be an alternative loss of function assay to performing it in the more biologically relevant cell lines.
Response: Axl and Axl-variants were essayed for transient overexpression in 293T cells by three different operators, resulting in all cases in failure of surface expression in 293T cells. To clarify this, lines 154-5 have been modified and expression of Axl variants in transfected 293T cells is now included in Fig. S1C.
- Question / Comment: A number of figures only report relative values when raw values are also needed to confirm that the data lies within the functional range of the assay- these figures include 1C, 1E, 1G, 2B-D, 2H, 4A-B, 5B-D and 7.
Response: Following Reviewer’s comment, we have included as supplementary figures the raw values obtained for each particular experiment. Because of this comment, we found that some comparative analyses were performed between experiments with different numbers of infected cells per well, resulting in significantly different number of infected cells and biasing the possible comparison. To solve this issue, we have performed additional experimentation, with infections in different cell lines carried out in parallel and achieving similar number of infected cells in the different cell lines, ensuring reliable comparisons. In consequence, Figs. 5H and Fig 7 have been revised.
- Question / Comment: In figures 2B-D, the use of ribavirin doesn’t appear to act as an appropriate control of post-entry steps for R428 and EIPA since ribavirin inhibited entry at all time points (instead of only at later timepoints) and it inhibited Lassa entry at significantly higher levels than R428 and EIPA. An alternative drug may need to be used as a better control, or ribavirin may need to be given at later time points or in smaller amounts to counteract longer-lasting effects
Response: We agree with the Reviewer, and to address this point, we have now included a new supplementary figure (Fig. S2C). Fig. S2C shows that Rib is not able to reduce LASV infectivity when present only during viral entry (Entry assay). In the main text, lines 222-3 were modified to clarify this point.
- Question / Comment: In figure 2E, the authors should justify using the TCEP biotin assay to assess virus internalization over imaging or determining viral titer after leaving the cells on ice, which would be more quantifiable. In fact in reference 69, the authors using this same assay specify that they left the cells on ice both during incubation with the virus and with incubation with TCEP, which allowed for a timed visualization of cell entry. This timed visualization isn’t possible in 2F since biotin signal from internalized virus could be seen in the DMSO treatment condition 0 minutes after TCEP quenching.
Response: Contrary to confocal visualization or viral titration, the TCEP assay is able to discern between virus present at “open cups” and closed vesicles still in the membrane proximity. Therefore, the TCEP assay constitutes a more sensitive approach than aforementioned techniques. In Fig 2E, signal from GP2 is only visible at 0 min after infection in absence of TCEP (used as infection control), but is undetectable in presence of TCEP at that time point. To further clarify the experimental set up, Fig. 2E and lines 232-4 in the manuscript were modified.
- Question / Comment: In figures 3A and B, Pearson’s coefficient may not be the best assessment of NP colocalization since the DG and Axl signals are significantly greater and more diffuse than the NP signal. This is reflected in the moderate difference in Pearson’s coefficient between 3A and 3B. A better assessment may be NP-DG or Axl double-positive puncta, which appears to be the biggest difference between 3A and 3B.
Response: We agree with the Reviewer and quantifications of co-localization of Axl/NP (Fig. 3) and HGFR/NP (Fig. 5) are now measured by double-positive puncta. Lines 659-62 (Material and methods’ section) in the manuscript were modified to add the method used.
- Question / Comment: In figure 5D, the authors could include a control to indicate that EMD truly has an additive effect specific with R428 as entry inhibitors as compared to any anti-Lassa drug
Response: We again agree with the Reviewer and combinatorial experiments with different drugs could be envisioned. Nevertheless, the limited antiviral drugs catalogue against mammarenavirus infection restricts the potential controls to use here. As both RTKs, Axl and HGFR, participate in macropinocytosis, it was conceivable that their inhibition by R428 and EMD, would not added their effects due to overlapped affected cellular mechanism. Therefore, we performed the experiments presented in Fig. 5H. To further elucidate the interaction between both drugs, we performed extended analysis to determine the type of interaction between both drugs (Tables 1 and 2).
- Question / Comment: In figure 7, it seems like there may be more use in varying the concentration of R248 at more points compared to ribavirin since it has a lower concentration of use and therefore might be more viable as a drug candidate. It would also be more likely to generate differences than two points at a 2x dilution.
Response: Following the Reviewer’s comment, we extended the conditions tested in Fig. 7, covering more Rib and R428 concentrations, and thus providing the antiviral effect of a wider catalogue of possible drug combinations. Is noteworthy that, because a previous comment from the Reviewer, this figure has been revised. Current revised Fig. 7 covers the combinations of Rib (0, 2.5, 5, 10 and 20 µM), EMD (5 µM) and R428 (0, 0.125, 0.25, 0.5 and 1µM). Furthermore, as done for Fig. 5H, we analysed the drug interaction observed in Fig 7 (Tables 3 and 4), and found synergistic effect at all concentrations tested in A549 and in moderate doses of R428 in HT-1080. Revised results are now commented in lines 440-53.
Minor points
- Question / Comment: On page 2 line 93, the authors should explain in more detail the concept of “apoptotic mimicry” and how that relates to viral entry.
Response: Following Reviewer’s comment, introductory sentences are introduced in lines 92-7 of the text.
- Question / Comment: On page 3 line 101, the authors should briefly explain the context that Axl was thought to co-operate with DG and how under-glycosylation of DG affected the interaction of Axl and DG.
Response: Lines 101-3 have been modifying accordingly to Reviewer’s comment.
- Question / Comment: On page 4 line 159, the authors should provide a citation if Mer/Tyro3 upregulation has been seen following Axl inhibition.
Response: Although no compensation is reported in previous literature, we conceived that Axl RTK depletion could possibly lead to compensation by increased expression of the other TAM family members, Mer or Tyro3. To avoid misinterpretations, we modified the sentence at line 159-63.
- Question / Comment: On page 4 line 171, the authors should explain why an Ebola virus construct was necessary to infect 293T cells.
Response: Lines 178-9 have been modified to include the reason EBOV was used as a control in these experiments.
- Question / Comment: On page 8 line 246, it can’t be determined whether DG clusters are truly “preformed” prior to interaction with LCMV/LASV- the word should be removed.
Response: “Preformed” has been removed from sentence.
- Question / Comment: On page 10 line 279, it should also be noted that R248 appeared to induce an IFNB expression in mock infected HT-1080 cells, possibly making R248 in HT-1080 cells more complicated to study viral replication than in other cell type
Response: We agree with the Reviewer, and commented this in lines 309-11.
- Question / Comment: On page 10 line 307, it may be misleading to say that HGF significantly promoted infection with a 40% increase. The authors should also address why the timing of HGF appeared to be more specific than the drug treatments used throughout the manuscript.
Response: In agreement with Reviewer’s comment, we changed the text in line 339-41 to avoid any kind of misinterpretation.
To address effect of HGF on HGFR activation over time, we performed additional experimentation and monitored the phosphorylation of HGFR upon HGF addition. Our results (Fig. 5D) show that phosphorylated HGFR signal is already reduced as early as 30 minutes after HGF addition, becoming undetectable 1h after addition, in line with previous results (1). We commented on this in lines 335-8 and new results are included in Fig. 5D.
- Question / Comment: On page 13 line 351, EIPA and R248 also seemed to have NP puncta in proximity to the plasma membrane- the authors should explain any differences they noted with EMD compared to these two drugs in this regard.
Response: Following Reviewer’s comment, lines 426-8 were modified to comment the differences of the effect of EIPA, R428 and EMD on the traffic of dextran- and virus-containing vesicles.
- Question / Comment: On page 14 line 410, the “land and stick” usage of words didn’t make sense for the model being described.
Response: The line 497 has been modified according to Reviewer’s comment.
- Question / Comment: On page 15 line 433, the results in this manuscript aren’t necessarily in contradiction with earlier results since the earlier results were performed in 293T cells expressing highly glycosylated DG- DG might take precedence over Axl and HGFR.
Response: As pointed out by the Reviewer, in (2), researchers extensively used 293T cells, which, as A549 cells, express highly glycosylated DG, to demonstrate the effect of Axl RTK in LASV entry. In our case, we investigated LASV entry in both contexts, highly-glycosylated and non-glycosylated DG, using A549 and HT-1080 cells, respectively. We found that, in both cell lines, inhibition of Axl RTK resulted in LASV entry inhibition, demonstrating that the aforementioned host factor participates in viral entry. Therefore, the DG glycosylation extent in both cell lines, 293T and A549 seems to be comparable, rendering a similar context for LASV entry.
Nevertheless, in addition to the differences caused by the LASV surrogate used in both studies (VSV-based pseudovirus platform or recombinant LCMV), it is possible that the ectopic expression of Axl RTK in 293T cells (2) does not completely recapitulate the cellular factor functionality. To reflect this, lines 523-7 and 537-9 have been modified in the manuscript.
Submission Date
15 June 2020
Date of this review
23 Jun 2020 21:00:25
REFERENCES:
- Boccaccio C, Ando M, Comoglio PM. A differentiation switch for genetically modified hepatocytes. FASEB J. 2002;16(1):120-2.
- Brouillette RB, Phillips EK, Patel R, Mahauad-Fernandez W, Moller-Tank S, Rogers KJ, et al. TIM-1 Mediates Dystroglycan-Independent Entry of Lassa Virus. J Virol. 2018;92(16).

Reviewer 2 Report
In this manuscript, Fedeli and colleagues investigated the contribution of the two LASV receptors Axl and HGFR in virus entry. Major findings are that both host receptors participate in the macropinocytosis-related LASV entry process and that their inhibition by chemical compounds results in a significant antiviral effect. Overall, the manuscript is well written and the results are of interest. The experimental work is clearly presented and technically sound. However, I have some concerns that need to be clarified prior publication.
Major comments:
Fig. 1A shows the cell surface expression of Axl in A549 cells and HT-1080 cells. These results were already published by the same group (see Fig. 3F in Fedeli et al., JVI 2018 Axl Can Serve as Entry Factor for Lassa Virus Depending on the Functional Glycosylation of Dystroglycan). The authors should comment on the added value to present published data in the present work.
Similarly, Fig. 2C shows the dose- and time-dependent inhibitory effect of the chemical compounds R428, EIPA and ribavirin on the viral replication of rLCMV/LASVGP in HT-1080 cells. Same results are shown for R428 and ribavirin using rLCMV/LASVGP and HT-1080 cells in Fig. 6D in Fedeli et al., JVI 2018. The graph for R428 in Fig. 2C (current manuscript) and Fig. 6D (Fedeli et al., JVI 2018) looks virtually identical. Did the authors re-use previously published findings in the current manuscript? Please comment.
Fig. 1C shows the inhibitory effect of R428 on viral replication of rLCMV/LASVGP in A549, HT-1080 and HUVEC cells. Using an inhibitor concentration ranging from 1-5 µM, the authors observed an almost complete inhibition of viral replication at a concentration of 3 µM in HT-1080 cells. This contrasts previous findings (Fig. 6D, Fedeli et al., JVI 2018), demonstrating an almost complete reduction of viral replication in HT-1080 cells when using a significant lower concentration of R428. Did the authors use different infection doses, which may explain the observed differences between the two studies?
Fig. 3B. Immunofluorescence studies failed to demonstrate a co-localization between LCMV NP and Axl, suggesting that the interaction may not occur at the cell surface. Did the authors look for an interaction between NP and Axl in the very early steps of endocytosis? Why did the authors not perform similar co-localization studies for HGFR and LCMV NP?
Is the molecular mechanism known how R428-induced inhibition of Axl prevents the entry of LASV? Is the catalytic activity of Axl targeted by R428 required for LASV entry? Does inhibiting the catalytic activity impact cell surface expression of Axl?
Fig. 5B demonstrates that inhibition of HGFR by EMD treatment reduced rLCMV/LASVGP entry in a dose-dependent manner in A549 and HT-1080 cells, with a stronger antiviral effect observed in HT-1080 cells. Co-treatment studies using EMD and R428 showed that EMD significantly enhanced the antiviral effect of R428 (Fig. 5D). Interestingly, the additive effect of EMD was more pronounced in A549 cells than in HT-1080 cells. Please comment.
Minor comments:
Lines 257, 344 and 348 (figure legends): Change µM to µm as you refer to the scale bar.
Line 271: „type I IFN-I signaling “. Please use either type I IFN signaling or IFN-I signaling.
Lines 522 and 529: 48h vs. 1 h
Lines 527 and 528: 0.1 % vs. 5%
Lines 527 and 567: (wt/vol) vs. v/v
Figs. 1, 5 and S3: Delete additional full stop at the end of the figure legends.
Author Response
Dear Sir, dear Madam,
Please, find below the addressed comments from Reviewers’ revision of the manuscript entitled « ROLE OF RECEPTOR TYROSINE KINASES IN LASSA VIRUS CELL ENTRY".
We appreciate very much the critical and constructive comments from all Reviewers. In addition to the indicated modifications to specifically address the Reviewer’s comments, we have also included other changes in the manuscript to make a clearer description of our findings.
Sincerely yours,
Héctor Moreno, PhD.
Lausanne, 31.7.2020.
REVIEWER 2
Open Review
(x) I would not like to sign my review report
( ) I would like to sign my review report
English language and style
( ) Extensive editing of English language and style required
( ) Moderate English changes required
(x) English language and style are fine/minor spell check required
( ) I don't feel qualified to judge about the English language and style
|
Yes |
Can be improved |
Must be improved |
Not applicable |
|
|
Does the introduction provide sufficient background and include all relevant references? |
(x) |
( ) |
( ) |
( ) |
|
Is the research design appropriate? |
(x) |
( ) |
( ) |
( ) |
|
Are the methods adequately described? |
(x) |
( ) |
( ) |
( ) |
|
Are the results clearly presented? |
(x) |
( ) |
( ) |
( ) |
|
Are the conclusions supported by the results? |
(x) |
( ) |
( ) |
( ) |
Comments and Suggestions for Authors
In this manuscript, Fedeli and colleagues investigated the contribution of the two LASV receptors Axl and HGFR in virus entry. Major findings are that both host receptors participate in the macropinocytosis-related LASV entry process and that their inhibition by chemical compounds results in a significant antiviral effect. Overall, the manuscript is well written and the results are of interest. The experimental work is clearly presented and technically sound. However, I have some concerns that need to be clarified prior publication.
Major comments:
- Question / Comment: 1A shows the cell surface expression of Axl in A549 cells and HT-1080 cells. These results were already published by the same group (see Fig. 3F in Fedeli et al., JVI 2018 Axl Can Serve as Entry Factor for Lassa Virus Depending on the Functional Glycosylation of Dystroglycan). The authors should comment on the added value to present published data in the present work.
Response: In agreement with Reviewer’s comment, we have moved aforementioned panels of Fig 1A to Fig. S1A-B and cited (1) instead. To clarify that the presented results are just a validation of previous studies; lines 147 and 152 were modified accordingly.
- Question / Comment: Similarly, Fig. 2C shows the dose- and time-dependent inhibitory effect of the chemical compounds R428, EIPA and ribavirin on the viral replication of rLCMV/LASVGP in HT-1080 cells. Same results are shown for R428 and ribavirin using rLCMV/LASVGP and HT-1080 cells in Fig. 6D in Fedeli et al., JVI 2018. The graph for R428 in Fig. 2C (current manuscript) and Fig. 6D (Fedeli et al., JVI 2018) looks virtually identical. Did the authors re-use previously published findings in the current manuscript? Please comment.
Response: The reviewer is correct and we apologize for the mistake. We have removed the Fig. 2C from the current manuscript and cited the previous publication (1) instead. Line 221 was modified to clarify this.
- Question / Comment: 1C shows the inhibitory effect of R428 on viral replication of rLCMV/LASVGP in A549, HT-1080 and HUVEC cells. Using an inhibitor concentration ranging from 1-5 µM, the authors observed an almost complete inhibition of viral replication at a concentration of 3 µM in HT-1080 cells. This contrasts previous findings (Fig. 6D, Fedeli et al., JVI 2018), demonstrating an almost complete reduction of viral replication in HT-1080 cells when using a significant lower concentration of R428. Did the authors use different infection doses, which may explain the observed differences between the two studies?
Response: The X-axis of Fig. 1D was labelled wrong and the current figure is now corrected.
- Question / Comment: 3B. Immunofluorescence studies failed to demonstrate a co-localization between LCMV NP and Axl, suggesting that the interaction may not occur at the cell surface. Did the authors look for an interaction between NP and Axl in the very early steps of endocytosis?
Response: Following Reviewer’s comment, we have performed additional experimentation and measured NP/Axl RTK co-localization during the first 10 minutes upon viral infection (Revised Fig. 3B). To comment on these results, lines 275-8 were modified.
- Question / Comment: Why did the authors not perform similar co-localization studies for HGFR and LCMV NP?
Response: Following the Reviewer’s comment, we performed extended experimentation and followed up NP/HGFR colocalization over early times upon attachment. The new results are now included in Fig. 5D. We commented these results in lines 374-7.
- Question / Comment: Is the molecular mechanism known how R428-induced inhibition of Axl prevents the entry of LASV?
Response: The mechanism of action of R428 on the entry pathway of LASV is currently unknown. In the present study, we provide evidence that Axl-RTK inhibition dramatically affected classical macropinocytosis. Given that LASV enter target cells through a macropinocytosis-related pathway, it is strongly suggested that Axl RTK signalling, required for maintaining a basal level of macropinosome traffic is required for LASV entry.
- Question / Comment: Is the catalytic activity of Axl targeted by R428 required for LASV entry?
Response: As R428 prevents Axl RTK phosphorylation and activation (2), it is most likely that Axl catalytic activity is required for LASV entry. This is supported by the increased sensitivity of HT-1080 cells to LASV infection, on which, due to the absence of glycosylated DG, Axl RTK acts as a receptor for LASV attachment (3) and its signalling is required for subsequent internalization. This is now commented in lines 504-6.
- Question / Comment: Does inhibiting the catalytic activity impact cell surface expression of Axl?
Response: Following Reviewer’s comment, we performed additional experimentation to address the impact of R428 or EMD treatment on Axl and HGFR RTKs, respectively (Figs. 1B and 5B). Lines 159 and 330 have been modified to include the comments on this results.
- Question / Comment: 5B demonstrates that inhibition of HGFR by EMD treatment reduced rLCMV/LASVGP entry in a dose-dependent manner in A549 and HT-1080 cells, with a stronger antiviral effect observed in HT-1080 cells. Co-treatment studies using EMD and R428 showed that EMD significantly enhanced the antiviral effect of R428 (Fig. 5D). Interestingly, the additive effect of EMD was more pronounced in A549 cells than in HT-1080 cells. Please comment.
Response: We agree with the Reviewer’s comment and performed additional experimentation and analysis. We found that comparisons between A549 and HT-1080 cells were done with experiments that did not have comparable number of infected cells as reference, leading to important misinterpretations of the results. To solve this issue, we have performed additional experimentation, performing infections in both cell lines in parallel and comparing dataset with comparable number of infected cells per well. To show this, raw data is included in corresponding supplementary figures.
The revised results show that the combinatorial effect of EMD and R428 (Fig. 5H) and Rib (Fig. 7) exert a more potent inhibition in rLCMV/LASVGP infection in A549 cells than in HT1080 cells. A comment in this regard is now included in lines 382-7 and 447-53.
Minor comments:
- Question / Comment: Lines 257, 344 and 348 (figure legends): Change µM to µm as you refer to the scale bar.
Response: Mentioned lines have been modified.
- Question / Comment: Line 271: „type I IFN-I signaling “. Please use either type I IFN signaling or IFN-I signaling.
Response: The mentioned misspelling has been corrected.
- Question / Comment: Lines 522 and 529: 48h vs. 1 h
Response: The misspelling has been corrected, and extended to the rest of the document.
- Question / Comment: Lines 527 and 528: 0.1 % vs. 5%
Response: The misspelling has been corrected, and extended to the rest of the document.
- Question / Comment: Lines 527 and 567: (wt/vol) vs. v/v
Response: The misspelling has been corrected (Lines 628 and 667).
- Question / Comment: 1, 5 and S3: Delete additional full stop at the end of the figure legends.
Response: Mentioned errors have been corrected.
Submission Date
15 June 2020
Date of this review
25 Jun 2020 20:18:23
REFERENCES:
- Fedeli C, Torriani G, Galan-Navarro C, Moraz ML, Moreno H, Gerold G, et al. Axl Can Serve as Entry Factor for Lassa Virus Depending on the Functional Glycosylation of Dystroglycan. J Virol. 2018;92(5).
- Holland SJ, Pan A, Franci C, Hu Y, Chang B, Li W, et al. R428, a selective small molecule inhibitor of Axl kinase, blocks tumor spread and prolongs survival in models of metastatic breast cancer. Cancer Res. 2010;70(4):1544-54.
- Oppliger J, Torriani G, Herrador A, Kunz S. Lassa Virus Cell Entry via Dystroglycan Involves an Unusual Pathway of Macropinocytosis. J Virol. 2016;90(14):6412-29.

Reviewer 3 Report
The manuscript “Role of receptor tyrosine kinases in Lassa virus cell entry” by Fedeli et al describes experiments that furthers our understanding of Lassa virus entry. This manuscript follows up on an earlier report that implicates both HGFR and micropinocytosis during early stages of cellular infection by Lassa virus. Their findings substantially contribute to our understanding of Lassa virus entry and identify potential therapeutic targets for a disease with no approved treatments. As is typical of the Kunz laboratory, their studies are well controlled, the manuscript clearly written and experiments presented in a logical manner. This manuscript should be of great interested to anyone in the arenavirus field and those studying viral entry.
Major Comment:
While the authors discussed and described experiments concerning the interferon response on a cellular level, it would be interesting for the authors to comment on how blocking a negative regulator of IFN might affect the immune response as a whole. Similarly, since HGFR is important in tissue regeneration and function, and Lassa fever is known to impair liver function, it would be prudent to briefly discuss the implications of blocking this pathway to liver function during LF as well.
Author Response
Dear Sir, dear Madam,
Please, find below the addressed comments from Reviewers’ revision of the manuscript entitled « ROLE OF RECEPTOR TYROSINE KINASES IN LASSA VIRUS CELL ENTRY".
We appreciate very much the critical and constructive comments from all Reviewers. In addition to the indicated modifications to specifically address the Reviewer’s comments, we have also included other changes in the manuscript to make a clearer description of our findings.
Sincerely yours,
Héctor Moreno, PhD.
Lausanne, 31.7.2020.
Reviewer 3.
Open Review
( ) I would not like to sign my review report
(x) I would like to sign my review report
English language and style
( ) Extensive editing of English language and style required
( ) Moderate English changes required
(x) English language and style are fine/minor spell check required
( ) I don't feel qualified to judge about the English language and style
|
Yes |
Can be improved |
Must be improved |
Not applicable |
|
|
Does the introduction provide sufficient background and include all relevant references? |
(x) |
( ) |
( ) |
( ) |
|
Is the research design appropriate? |
(x) |
( ) |
( ) |
( ) |
|
Are the methods adequately described? |
(x) |
( ) |
( ) |
( ) |
|
Are the results clearly presented? |
(x) |
( ) |
( ) |
( ) |
|
Are the conclusions supported by the results? |
(x) |
( ) |
( ) |
( ) |
Comments and Suggestions for Authors
The manuscript “Role of receptor tyrosine kinases in Lassa virus cell entry” by Fedeli et al describes experiments that furthers our understanding of Lassa virus entry. This manuscript follows up on an earlier report that implicates both HGFR and micropinocytosis during early stages of cellular infection by Lassa virus. Their findings substantially contribute to our understanding of Lassa virus entry and identify potential therapeutic targets for a disease with no approved treatments. As is typical of the Kunz laboratory, their studies are well controlled, the manuscript clearly written and experiments presented in a logical manner. This manuscript should be of great interested to anyone in the arenavirus field and those studying viral entry.
Major Comment:
- Question / Comment: While the authors discussed and described experiments concerning the interferon response on a cellular level, it would be interesting for the authors to comment on how blocking a negative regulator of IFN might affect the immune response as a whole. Similarly, since HGFR is important in tissue regeneration and function, and Lassa fever is known to impair liver function, it would be prudent to briefly discuss the implications of blocking this pathway to liver function during LF as well.
Response: In agreement with Reviewer, we have commented and discussed the consequences that inhibition of an IFN regulator such Axl RTK may have in the innate immune response (lines 543-9) as well as the effect of EMD on liver regeneration in the context of Lassa infection (lines 517-24).
Submission Date
15 June 2020

Reviewer 4 Report
Major comments:
- The manuscript needs to be edited for language. The results section should be more descriptive of the assays and how they were performed.
- Please discuss the possibility of off-target effects: R428 targets the Axl ATP binding site with an IC50 < 30 nM. For most of the assays in this manuscript a much higher concentration was used.
Minor comments:
Line 32: should be modified to: “The Old World (OW) mammarenavirus Lassa virus (LASV) is a zoonotic pathogen and the etiologic agent of Lassa hemorrhagic fever (LHF), a severe disease with high case fatality rate in humans”
Line 35: change LASV HF to LHF
Line 37: change “considering” to “because of”
Line 62: omit the “the” in ”…OW mammarenaviruses LASV and the lymphocytic..”
Line 94: change “playing” to ”and plays”
Line 97: font size of “GP” is off
Line 106: change “cell” to “cells”
Line 109: delete “envelope”
Line 115: change to ”…macropinocytosis, participate in entry of a range of viruses, and therefore might also be important for LASV entry.
Line 121: change “and inhibitor currently tested” to “an inhibitor currently tested”
Line 122-124: this sentence does not make sense. Please rephrase.
Line 126: should be: “expressing the envelope (GP) of LASV”
Line 148: How did the authors conclude that DG is under-glycosylated in this cell line? Couldn’t it be that it is just expressed at lower levels?
Line 153-154: Is the statement regarding the cytotoxicity of the mutants based on data not shown or a reference? Please indelicate in the manuscript.
Line 155: modify to “…with an IC50 < 30 nM in tissue culture conditions. It is a reversible inhibitor, and shows…”
Figure 1:
Figure 1B and 1C: please describe the assay in short in the results section. It is not clear how the assay was performed. Also, For how long were cells treated with R428? How much AdV5-EGFP/pseudotypes were added to cells? For how long overall were cells incubated with virus or pseudotypes or AdV? These details are also not in the methods section. What software was used for dose response analysis?
Figure 1E and 1G: Is the change in infection following Axl expression significant? please include statistics. Also, in the figure the Y axes are labeled fold-induction (Fig 1E) and relative entry (Fig 1G) whereas in the figure legend it’s relative entry (Fig 1E) and infection (Fig 1G). Pseudotype transduce cells whereas rLCMV/LASVGP and AdV5 infect cells. Appropriate labeling will therefore be “relative transduction/infection”
Figure 2:
Figure 2A: Has the timeline associated with each virus lifecycle event been previously established? If yes, please reference.
Fig 2F: Again, please include a short description of the assay in the results section. When are cells treated with DMSO/R428/EIPA and for how long? please use densitometry (using Image J.) to evaluate level of inhibition of internalization.
Line 217-219: Please describe shortly the assays and the drugs used.
Fig 3C: correct graph legend- “infected cells (10 min pi)” should be “infected cells”
Line 272-273: does not make sense. Please revise
Line 275-276: does not make sense. Please revise
Line 299: What is the IC50 of EMD against HGFR?
Figure 5D and 7: A more informative assay/analysis would have tested/determined if the compounds have additive or synergistic effect.
Author Response
Response to Reviewer #4
Dear Sir, dear Madam,
Please, find below the addressed comments from Reviewers’ revision of the manuscript entitled « ROLE OF RECEPTOR TYROSINE KINASES IN LASSA VIRUS CELL ENTRY".
We appreciate very much the critical and constructive comments from all Reviewers. In addition to the indicated modifications to specifically address the Reviewer’s comments, we have also included other changes in the manuscript to make a clearer description of our findings.
Sincerely yours,
Héctor Moreno, PhD.
Lausanne, 31.7.2020.
Open Review
(x) I would not like to sign my review report
( ) I would like to sign my review report
English language and style
( ) Extensive editing of English language and style required
(x) Moderate English changes required
( ) English language and style are fine/minor spell check required
( ) I don't feel qualified to judge about the English language and style
|
Yes |
Can be improved |
Must be improved |
Not applicable |
|
|
Does the introduction provide sufficient background and include all relevant references? |
(x) |
( ) |
( ) |
( ) |
|
Is the research design appropriate? |
( ) |
( ) |
( ) |
( ) |
|
Are the methods adequately described? |
( ) |
(x) |
( ) |
( ) |
|
Are the results clearly presented? |
( ) |
(x) |
( ) |
( ) |
|
Are the conclusions supported by the results? |
(x) |
( ) |
( ) |
( ) |
Comments and Suggestions for Authors
Major comments:
- Question / Comment: The manuscript needs to be edited for language. The results section should be more descriptive of the assays and how they were performed.
Response: Following Reviewer’s comment, we have revised the language of the whole manuscript.
Furthermore, we now provide further description of “Entry”, “post-entry”, “time of addition” and “TCEP” assays in lines 164-71, 210-17, 240-47, 328-31, 371-4 and 601-618.
- Question / Comment: Please discuss the possibility of off-target effects: R428 targets the Axl ATP binding site with an IC50 < 30 nM. For most of the assays in this manuscript a much higher concentration was used.
Response: Among the available Axl RTK inhibitors, R428 is the most specific and is extensively used for this purpose (1-4). Despite the IC50 of Axl phosphorylation in HeLa cells is indeed in the range of the nM scale (5), the IC50 values of viral entry inhibition are in the scale of µM (6, 7). Moreover, considering the possible additional targets of R428, Tie-2, Ftl-1, Flt-3, Ret, and Abl (5), to dissect Axl RTK participation in LASV entry, we choose a short time-window of drug action, combined with drug washout (8). To comment on this, we have modified lines 159-63 in the text.
Minor comments:
- Question / Comment: Line 32: should be modified to: “The Old World (OW) mammarenavirus Lassa virus (LASV) is a zoonotic pathogen and the etiologic agent of Lassa hemorrhagic fever (LHF), a severe disease with high case fatality rate in humans”
Response: The sentence has been modified according to Reviewer’s comment.
- Question / Comment: Line 35: change LASV HF to LHF
Response: The misspelling has been corrected.
- Question / Comment: Line 37: change “considering” to “because of”
Response: Following Reviewer’s comment, we changed the words for a better understanding.
- Question / Comment: Line 62: omit the “the” in ”…OW mammarenaviruses LASV and the lymphocytic..”
Response: “the” has been removed in the mentioned sentence.
- Question / Comment: Line 94: change “playing” to ”and plays”
Response: To address a previous Reviewer’s comment, the mentioned sentence has been removed. Therefore, this comment was not taken into account.
- Question / Comment: Line 97: font size of “GP” is off
Response: We changed the font format of “GP” in the mentioned sentence and extended the comment to the rest of the manuscript.
- Question / Comment: Line 106: change “cell” to “cells”
Response: The misspelling has been corrected in line 109.
- Question / Comment: Line 109: delete “envelope”
Response: “Envelope” has been removed from the indicated sentence.
- Question / Comment: Line 115: change to ”…macropinocytosis, participate in entry of a range of viruses, and therefore might also be important for LASV entry.
Response: As suggested by the Reviewer, we modified the mentioned sentence.
- Question / Comment: Line 121: change “and inhibitor currently tested” to “an inhibitor currently tested”
Response: The misspelling has been corrected.
- Question / Comment: Line 122-124: this sentence does not make sense. Please rephrase.
Response: This sentence has been removed from the manuscript.
- Question / Comment: Line 126: should be: “expressing the envelope (GP) of LASV”
Response: As indicated by the Reviewer, we modified the indicated sentence.
- Question / Comment: Line 148: How did the authors conclude that DG is under-glycosylated in this cell line? Couldn’t it be that it is just expressed at lower levels?
Response: In (9), the comparison of the ratios of βDG and glycosylated αDG demonstrate that while HUVEC cells have a comparable βDG expression to A549 or SAEC cells, the expression of. glycosylated αDG is notably reduced in HUVEC cells. To clarify this, we have included the aforementioned reference in the text (line 149).
- Question / Comment: Line 153-154: Is the statement regarding the cytotoxicity of the mutants based on data not shown or a reference? Please indelicate in the manuscript.
Response: Three different operators performed experimentation to transiently overexpress Axl RTK variants in 293T cells, but failed. We have clarified this in lines 154-6 and transient expression of Axl variants in 293T cells is now included in Fig. S1C.
- Question / Comment: Line 155: modify to “…with an IC50 < 30 nM in tissue culture conditions. It is a reversible inhibitor, and shows…”
Response: Lines 157-8 have been modified according to Reviewer’s comment.
- Question / Comment: Figure 1B and 1C: please describe the assay in short in the results section. It is not clear how the assay was performed. Also, For how long were cells treated with R428? How much AdV5-EGFP/pseudotypes were added to cells? For how long overall were cells incubated with virus or pseudotypes or AdV? These details are also not in the methods section. What software was used for dose response analysis?
Response: Following this and previous comment of the Reviewer, lines 164-71, 210-17, 240-47, 328-31, 371-4 and 601-618 have been modified to provide further description of how experimentation was performed. In addition, raw original data, including the number of positive cells per well for each experiment, is included as Supplementary figures for better interpretation.
The analyses of drug interactions in Figs 5H and 7 was performed with CompuSyn software (10). Lines 382-7 and 448-453 were modified to comment on this.
- Question / Comment: Figure 1E and 1G: Is the change in infection following Axl expression significant? please include statistics.
Response: Statistics are now included in the figures and lines 185, 188, and figure legends were modified accordingly.
- Question / Comment: Also, in the figure 1 the Y axes are labeled fold-induction (Fig 1E) and relative entry (Fig 1G) whereas in the figure legend it’s relative entry (Fig 1E) and infection (Fig 1G). Pseudotype transduce cells whereas rLCMV/LASVGP and AdV5 infect cells. Appropriate labelling will therefore be “relative transduction/infection”
Response: In agreement with the Reviewer’s comment, Y axis of current figures F and H and lines 184 and 187-8 have been accordingly modified.
- Question / Comment: Figure 2A: Has the timeline associated with each virus lifecycle event been previously established? If yes, please reference.
Response: The timeline of events during LASV attachment, endocytosis and release to cytosol is not precisely defined. Nevertheless, the complete viral cycle of mammarenaviruses is estimated in 6-8 hours (11), which served us for rough estimation of the different steps required to complete the entire virus lifecycle.
- Question / Comment: Fig 2F: Again, please include a short description of the assay in the results section. When are cells treated with DMSO/R428/EIPA and for how long? please use densitometry (using Image J.) to evaluate level of inhibition of internalization.
Response: Further descriptions of the experimental design are now included in corresponding Results section. Ratios of signal intensity (calculated by densitometry with ImageJ) are now displayed in Figure 2E.
- Question / Comment: Line 217-219: Please describe shortly the assays and the drugs used.
Response: This point has been already addressed in the Reviewer’s previous comment.
- Question / Comment: Fig 3C: correct graph legend- “infected cells (10 min pi)” should be “infected cells”
Response: The Fig. 3C has been corrected.
- Question / Comment: Line 272-273: does not make sense. Please revise
Response: According to the Reviewer’s comment, the aforementioned sentence (current lines 300-1) has been modified.
- Question / Comment: Line 275-276: does not make sense. Please revise
Response: Following Reviewer’s comment, we have modified the sentence in line 304-5.
- Question / Comment: Line 299: What is the IC50 of EMD against HGFR?
Response: EMD has an IC50 against HGFR phosphorylation of 3 nM (12), and is now indicated in line 330 of the manuscript.
- Question / Comment: Figure 5D and 7: A more informative assay/analysis would have tested/determined if the compounds have additive or synergistic effect.
Response: Following the Reviewer’s comment, we extended our study and performed analysis to determine the nature of the interaction between R428, EMD (Fig 5H) and Rib (Fig 7). For the analysis, we used CompuSyn softare (10). We commented the results in lines 382-7 and 448-453.
Submission Date
15 June 2020
Date of this review
29 Jun 2020 04:38:26
Final del formulario
© 1996-2020 MDPI (Basel, Switzerland) unless otherwise stated
REFERENCES:
- McDaniel NK, Iida M, Nickel KP, Longhurst CA, Fischbach SR, Rodems TS, et al. AXL Mediates Cetuximab and Radiation Resistance Through Tyrosine 821 and the c-ABL Kinase Pathway in Head and Neck Cancer. Clin Cancer Res. 2020.
- Myers SH, Brunton VG, Unciti-Broceta A. AXL Inhibitors in Cancer: A Medicinal Chemistry Perspective. J Med Chem. 2016;59(8):3593-608.
- Woo SM, Min KJ, Seo SU, Kim S, Kubatka P, Park JW, et al. Axl Inhibitor R428 Enhances TRAIL-Mediated Apoptosis Through Downregulation of c-FLIP and Survivin Expression in Renal Carcinoma. Int J Mol Sci. 2019;20(13).
- Umemura S, Sowa Y, Iizumi Y, Kitawaki J, Sakai T. Synergistic effect of the inhibitors of RAF/MEK and AXL on KRAS-mutated ovarian cancer cells with high AXL expression. Cancer Sci. 2020;111(6):2052-61.
- Holland SJ, Pan A, Franci C, Hu Y, Chang B, Li W, et al. R428, a selective small molecule inhibitor of Axl kinase, blocks tumor spread and prolongs survival in models of metastatic breast cancer. Cancer Res. 2010;70(4):1544-54.
- Meertens L, Labeau A, Dejarnac O, Cipriani S, Sinigaglia L, Bonnet-Madin L, et al. Axl Mediates ZIKA Virus Entry in Human Glial Cells and Modulates Innate Immune Responses. Cell Rep. 2017;18(2):324-33.
- Oppliger J, Torriani G, Herrador A, Kunz S. Lassa Virus Cell Entry via Dystroglycan Involves an Unusual Pathway of Macropinocytosis. J Virol. 2016;90(14):6412-29.
- Pasquato A, Fernandez AH, Kunz S. Studies of Lassa Virus Cell Entry. Methods Mol Biol. 2018;1604:135-55.
- Fedeli C, Torriani G, Galan-Navarro C, Moraz ML, Moreno H, Gerold G, et al. Axl Can Serve as Entry Factor for Lassa Virus Depending on the Functional Glycosylation of Dystroglycan. J Virol. 2018;92(5).
- Bijnsdorp IV, Giovannetti E, Peters GJ. Analysis of drug interactions. Methods Mol Biol. 2011;731:421-34.
- Moreno H, Tejero H, de la Torre JC, Domingo E, Martin V. Mutagenesis-mediated virus extinction: virus-dependent effect of viral load on sensitivity to lethal defection. PLoS One. 2012;7(3):e32550.
- Bladt F, Faden B, Friese-Hamim M, Knuehl C, Wilm C, Fittschen C, et al. EMD 1214063 and EMD 1204831 constitute a new class of potent and highly selective c-Met inhibitors. Clin Cancer Res. 2013;19(11):2941-51.

Round 2
Reviewer 1 Report
The authors have sufficiently addressed my previous comments. The only point I'll offer is that it appears from my end that not all of the supplemental figures appeared to be uploaded- I was only able to access supplemental figures 1-3 and some of those appeared to be different than the most recent manuscript. Updating the supplemental figures will be sufficient for acceptance.